# Optimizing Distributional Geometry Alignment with Optimal Transport for Generative Dataset Distillation

**Xiao Cui**[1,2]    **Yulei Qin**[3]    **Wengang Zhou**[1]    **Hongsheng Li**[2]    **Houqiang Li**[1]

[1] University of Science and Technology of China
[2] CUHK MMLab
[3] Independent Researcher

cuixiao2001@mail.ustc.edu.cn, yuleichin@126.com
{zhwg,lihq}@ustc.edu.cn, hsli@ee.cuhk.edu.hk

## Abstract

Dataset distillation seeks to synthesize a compact distilled dataset, enabling models trained on it to achieve performance comparable to models trained on the full dataset. Recent methods for large-scale datasets focus on matching global distributional statistics (e.g., mean and variance), but overlook critical instance-level characteristics and intraclass variations, leading to suboptimal generalization. We address this limitation by reformulating dataset distillation as an Optimal Transport (OT) distance minimization problem, enabling fine-grained alignment at both global and instance levels throughout the pipeline. OT offers a geometrically faithful framework for distribution matching. It effectively preserves local modes, intra-class patterns, and fine-grained variations that characterize the geometry of complex, high-dimensional distributions. Our method comprises three components tailored for preserving distributional geometry: (1) OT-guided diffusion sampling, which aligns latent distributions of real and distilled images; (2) label-image-aligned soft relabeling, which adapts label distributions based on the complexity of distilled image distributions; and (3) OT-based logit matching, which aligns the output of student models with soft-label distributions. Extensive experiments across diverse architectures and large-scale datasets demonstrate that our method consistently outperforms state-of-the-art approaches in an efficient manner, achieving at least 4% accuracy improvement under IPC=10 settings for each architecture on ImageNet-1K.

## 1    Introduction

The expansion of data has fueled advances in deep learning, but also introduced prohibitive costs in storage, computation, and energy [1, 2, 3]. To address these challenges, dataset distillation aims to synthesize a small set of training samples to expedite model training while maintaining comparable performance [4]. Such distillation not only improves accessibility and cost-efficiency, but also facilitates practical applications such as knowledge transfer [5], federated learning [6, 7], and continual learning [8, 9]. Moreover, it provides a valuable lens to investigate the theoretical principles underlying training efficiency and representation capacity in deep learning [10, 11].

Traditional dataset distillation methods can be broadly categorized into optimization-based [12, 13, 14, 15] and distribution-matching-based approaches [16, 17, 18, 19]. Despite their effectiveness, these methods remain largely restricted to small-scale, low-resolution datasets such as MNIST [20], CIFAR [21], or downsampled ImageNet subsets [22]. This limitation stems from the prohibitive

---

Corresponding authors: Wengang Zhou and Houqiang Li.

39th Conference on Neural Information Processing Systems (NeurIPS 2025).

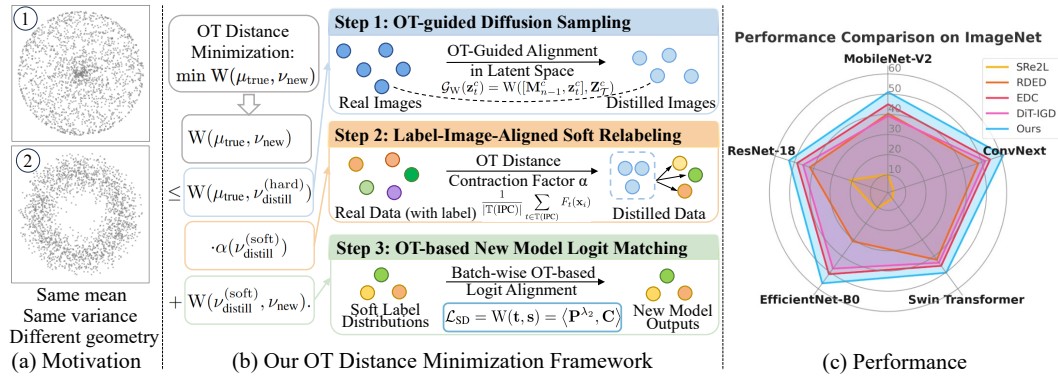

Figure 1: (a) Distributions with identical mean or variance may differ geometrically, causing biases in global-statistics-based optimization. (b) We decompose the OT objective into three stages: OT-guided diffusion sampling, label-image-aligned soft relabeling, and OT-based logit matching. (c) Our method consistently outperforms prior approaches across architectures on ImageNet-1K (IPC = 10).

computational cost of alternating optimization between the distilled data and the condensation model [23], and the reliance on integrated image representations that demand costly pixel-level refinement [24]. Recent efforts have explored generative and model-inversion-based techniques to overcome these scalability bottlenecks. Model-inversion-based methods [25, 26, 27, 28], originally proposed under data-free distillation framework [29, 30], rely entirely on global batch-normalization statistics extracted from pretrained models. While simple, this design imposes an inherent limitation: it fundamentally lacks the ability to recover or preserve instance-level, local distributional structures. In contrast, generative-model-based methods [24, 31, 32, 33] leverage real image samples during the sampling process, showing potential to approximate the true data distribution more faithfully.

However, existing generative approaches have yet to fully realize this promise, as they still solely focus on matching global gradient statistics. Besides, the fine-grained distributional structures are not properly captured by cosine-similarity-based diversity guidance, resulting in local mode collapse and distributional mismatch in the distilled set. To address this limitation, we propose a principled reformulation grounded in Optimal Transport (OT), which enables fine-grained distributional geometry alignment between real distribution and model output distribution. Specifically, we define *distributional geometry alignment* as preserving distribution-level global and local structures (e.g., coarse-grained patterns and subclass densities), rather than image-level features. Our key insight is that each real data point encapsulates rich intra-class semantic variation, such as the distinctive traits of different subclasses or local modes within the same class. OT inherently provides a geometrically faithful and perceptually aligned measure of distributional differences [34], making it especially promising for preserving and transferring these fine-grained semantic structures during distillation.

Building upon this, we formulate dataset distillation as an OT distance minimization problem. As shown in Figure 1, to make the alignment process tractable and optimization-friendly, we decompose the total OT distance into three complementary objectives that altogether contribute to its minimization: (1) instance-level transport in image latent space, (2) label-image alignment in label space, and (3) batch-wise logit alignment between new model predictions and soft targets. Such decomposition ensures alignment through the sequential stages of dataset distillation, ranging from image generation and label assignment to student model training. In the first stage, latent space transport is achieved by continuously computing the OT distance between the accumulated synthetic images (including the newly generated samples) and the real image batches at each sampling step. The gradients from this computation are used to guide the diffusion sampling process. In the second stage, we align the complexity of the synthetic image distribution with that of the soft label distribution, thereby narrowing the OT distance between the distilled data and the real data. In the final stage, we transfer the rich distributional geometric information embedded in the distilled set to new student models by minimizing the batch-wise OT distance between the student outputs and the soft-label distributions.

We evaluate our method across a diverse range of architectures, including ResNet, MobileNet, EfficientNet, Swin Transformer, ConvNet, and ConvNeXt. Our approach consistently outperforms

state-of-the-art methods across all datasets, architectures, and IPCs, achieving particularly strong results on ImageNet-1K [22] under the challenging IPC=10 scenario. Our contributions are threefold:

- We propose a novel perspective of dataset distillation by formulating the task as an OT distance minimization problem. We decompose the objective into three tractable components.

- We systematically enhance distributional geometry alignment through key stages of the pipeline, including image sampling, soft label relabeling, and student model logit matching.

- We demonstrate the effectiveness and generalizability of our method across a broad range of datasets and model architectures, which significantly surpass existing techniques.

## 2 Related Works

Numerous studies have investigated dataset distillation: initial works target low-resolution, small-scale datasets, while more recent methods address large-scale, higher-resolution scenarios.

### 2.1 Small-scale distillation methods

Traditional dataset distillation methods can be broadly classified into optimization-based and distribution-matching (DM)-based approaches. Optimization-based methods [12, 13, 35, 14, 15] adopt a bi-level optimization framework, where model parameters are updated in the outer loop while synthetic data are refined in the inner loop to match gradients or trajectories. In contrast, DM-based methods [16, 17, 18, 36, 37, 19] directly align the feature distributions of real and synthetic data, thereby avoiding costly nested optimization. However, all these methods exhibit high model dependence on the condensation model, which limits the versatility of the distilled datasets in generalizing across different architectures [38, 39]. Also, they incur significant time and memory costs due to three factors: (1) treating synthetic data as fixed entities, (2) requiring exhaustive pixel-level refinements, and (3) relying on real data for image refinement. As a result, traditional dataset distillation approaches are predominantly applied to small-scale datasets.

### 2.2 Large-scale distillation methods

Recent methods for large-scale, high-resolution datasets fall into two main categories: model-inversion-based and generative-model-based methods. Model-inversion-based methods [27, 28, 23, 40, 41, 42, 43] compress the real dataset into a compact model representation, eliminating the need for real data during image refinement. This reduces memory overhead and allows scaling to large datasets such as ImageNet-1K [22]. However, the lack of real data in the reconstruction process results in the loss of fine-grained instance-level information, which hinders the distilled dataset from accurately capturing the structural properties and instance-specific characteristics of the real distribution. Generative-model-based methods [32, 44, 45, 46, 24] leverage pretrained generative models to avoid pixel-level refinements. While generating one sample at a time reduces memory overhead and avoids treating data as fixed entities, independently synthesizing each sample prevents the distilled dataset from maintaining a coherent overall distribution, thereby limiting its ability to capture the full diversity and structural relationships of the real data distribution. Due to their inherent exclusion of real data during reconstruction, model-inversion-based methods fail to preserve fine-grained structures of the real distribution; accordingly, we adopt the generative-model-based paradigm as our starting point. We address the shortcomings of both families by proposing an OT framework that ensures distributional geometry alignment throughout the distillation process.

## 3 Preliminaries

Given images $\mathbf{x} \sim q(\mathbf{x})$, define the induced latent distribution $q_Z$ by $\mathbf{z}_0 = E(\mathbf{x})$, $\mathbf{z}_0 \sim q_Z(\mathbf{z_0})$, where $E$ is the encoder mapping images into latent space. A latent diffusion model learns $p_\phi(\mathbf{z}_0) \approx q_Z(\mathbf{z}_0)$, from which we can efficiently sample. Let $D$ be a decoder that reconstructs images via $\hat{\mathbf{x}} = D(\mathbf{z}_0)$. The forward noising process corrupts the clean latent $\mathbf{z}_0$ via Gaussian perturbations:

$$\mathbf{z}_t = \sqrt{\alpha_t}\mathbf{z}_0 + \sqrt{1 - \alpha_t}\boldsymbol{\epsilon}, \quad \boldsymbol{\epsilon} \sim \mathcal{N}(0, \mathbf{I}), \tag{1}$$

Table 1: Distributions involved in dataset distillation. All are defined over the joint space $\mathcal{X} \times \mathcal{Y}$, where $\mathcal{X}$ denotes the image space and $\mathcal{Y}$ denotes the label space.

| Distribution | Image Source | Label Source | Description |
|---|---|---|---|
| $\mu_{\text{true}}(\mathbf{x}, \mathbf{y})$ | Real (full) Images | Ground-truth $y(\mathbf{x})$ | True data-label distribution |
| $\nu_{\text{distill}}^{(\text{soft})}(\mathbf{x}, \mathbf{y})$ | Distilled (generated) Images | Teacher soft label $\mathbf{t}(\mathbf{x})$ | Distilled data with soft label |
| $\nu_{\text{distill}}^{(\text{hard})}(\mathbf{x}, \mathbf{y})$ | Distilled (generated) Images | One-hot label $y_{\text{onehot}}(\mathbf{x})$ | Distilled data with hard label |
| $\nu_{\text{new}}(\mathbf{x}, \mathbf{y})$ | Distilled (generated) Images | Student logit output $\mathbf{s}(\mathbf{x})$ | Output of the model trained on $\mathcal{S}$ |

where $\alpha_t$ controls the noise schedule. The reverse process reconstructs clean samples via a parameterized denoising function $\epsilon_\phi(\mathbf{z}_t, t)$, iteratively refining noisy inputs with update function $s$:

$$\mathbf{z}_{t-1} = s(\mathbf{z}_t, t, \epsilon_\phi(\mathbf{z}_t, t)). \tag{2}$$

Guided diffusion [47] modifies this process by introducing an auxiliary guidance function $\mathcal{G}(\mathbf{z}_t, t)$ that adjusts the sampling trajectory. This allows generation to be conditioned on labels, structural priors, or more abstract objectives by modifying $s(\mathbf{z}_t, t, \epsilon_\phi)$ to optimize for a task-specific constraint.

Influence-Guided Diffusion (IGD) [24] leverages guided diffusion for dataset distillation by modifying the reverse sampling process to generate training-optimal data. Instead of passively sampling from $p_\phi(\mathbf{x})$, IGD introduces trajectory influence function and diversity function into the diffusion process to prioritize samples. Given a guided diffusion framework, IGD modifies the sampling update as:

$$\mathbf{z}_{t-1} = s(\mathbf{z}_t, t, \epsilon_\phi) - \rho_t \nabla_{\mathbf{z}_t} \mathcal{G}_I(\mathbf{z_t}, t) - \gamma_t \nabla_{\mathbf{z}_t} \mathcal{G}_D(\mathbf{z}_t), \tag{3}$$

where $\mathcal{G}_I(\mathbf{z}_t, t)$ represents the influence function for global distributional trajectory matching, and $\mathcal{G}_D(\mathbf{z}_t)$ enforces diversity to prevent redundancy in the distilled dataset.

## 4 Methods

### 4.1 Problem Statement

Dataset distillation aims to construct a compact distilled dataset $\mathcal{S} \equiv \nu_{\text{distill}}(\mathbf{x}, \mathbf{y})$ (this means that $\nu_{\text{distill}}$ denotes the empirical distribution over dataset $\mathcal{S}$) from a real, full dataset $\mathcal{T} \equiv \mu_{\text{true}}(\mathbf{x}, \mathbf{y})$, such that $|\mathcal{S}| \ll |\mathcal{T}|$. A student model trained on $\mathcal{S}$ should mimic the performance of training on $\mathcal{T}$, i.e., the output distribution $\nu_{\text{new}}(\mathbf{x}, \mathbf{y})$ of the student model should remain close to the ground-truth distribution $\mu_{\text{true}}(\mathbf{x}, \mathbf{y})$. We formulate the dataset distillation objective as an OT minimization problem, where we minimize the Wasserstein distance $\text{W}(\mu_{\text{true}}, \nu_{\text{new}})$ between the ground-truth and student-induced distributions. The key distributions involved in the formulations are summarized in Table 1. We provide a detailed description of symbols in Appendix D.

### 4.2 Reconstructing the Optimal Transport Distance

We now provide a theoretical decomposition of our objective, $\text{W}(\mu_{\text{true}}, \nu_{\text{new}})$, by introducing two key principles: (1) the triangle inequality partitioning the discrepancy introduced before and after distilled set construction, and (2) a multiplicative contraction term reflecting the benefit of soft labels over hard labels. Unlike the commonly used measures such as KL divergence and cosine similarity, which do not satisfy the triangle inequality, the Wasserstein distance $\text{W}(\cdot, \cdot)$ is a proper metric on the space of distributions. This property allows us to decompose the total discrepancy as:

$$\text{W}(\mu_{\text{true}}, \nu_{\text{new}}) \leq \text{W}(\mu_{\text{true}}, \nu_{\text{distill}}^{(\text{hard})}) + \text{W}(\nu_{\text{distill}}^{(\text{hard})}, \nu_{\text{new}}). \tag{4}$$

As soft labels better approximate the true label distribution than one-hot assignments, recent works adopt soft labels instead of hard one-hot assignments, leading to the following relaxed upper bound:

$$\text{W}(\mu_{\text{true}}, \nu_{\text{new}}) \leq \underbrace{\text{W}(\mu_{\text{true}}, \nu_{\text{distill}}^{(\text{soft})})}_{\text{Dataset Discrepancy}} + \underbrace{\text{W}(\nu_{\text{distill}}^{(\text{soft})}, \nu_{\text{new}})}_{\text{Logit Matching Error}}. \tag{5}$$

The first term captures the mismatch between the distilled and real data distribution. The second term reflects the logit-wise alignment between the student model's output distribution and the soft labels

of the distilled data. Directly minimizing the first term is challenging due to too many variables to optimize. To analyze it further, we model the soft label advantage via a multiplicative relation:

$$W(\mu_{\text{true}}, \nu_{\text{distill}}^{(\text{soft})}) = W(\mu_{\text{true}}, \nu_{\text{distill}}^{(\text{hard})}) \cdot \alpha(\nu_{\text{distill}}^{(\text{soft})}) \tag{6}$$

The contraction factor $\alpha(\nu_{\text{distill}}^{(\text{soft})})$ measures how much soft labels reduce the discrepancy between label and image distributions compared to hard labels alone. The contraction is achieved by matching the complexity of the teacher-provided soft label distributions to that of the distilled image distribution. Since both $\nu_{\text{distill}}^{(\text{hard})}$ and $\mu_{\text{true}}$ use one-hot (hard) labels, their Wasserstein distance can be computed class-wise, by independently solving optimal transport between images of the same category:

$$W(\mu_{\text{true}}, \nu_{\text{distill}}^{(\text{hard})}) = \mathbb{E}_y \left[ W(\mu_{\text{true}}(\mathbf{x} \mid y), \nu_{\text{distill}}^{(\text{hard})}(\mathbf{x} \mid y)) \right] \tag{7}$$

where $\mathbb{E}_y$ denotes the expectation over label classes, which measures the average conditional Wasserstein distance across classes. Putting everything together, we arrive at a structured upper bound:

$$W(\mu_{\text{true}}, \nu_{\text{new}}) \leq \mathbb{E}_y \left[ W(\mu_{\text{true}}(\mathbf{x} \mid y), \nu_{\text{distill}}^{(\text{hard})}(\mathbf{x} \mid y)) \right] \cdot \alpha(\nu_{\text{distill}}^{(\text{soft})}) + W(\nu_{\text{distill}}^{(\text{soft})}, \nu_{\text{new}}) \tag{8}$$

where each term is controlled by a distinct design choice: OT-guided diffusion sampling, label-image-aligned soft label relabeling, and OT-based logit matching between the student model and the distilled dataset. This decomposition allows a principled basis of our method, which explicitly targets at minimization of each component. Our pseudocodes are provided in Appendix E.

### 4.3 OT-guided Diffusion Sampling (OTG)

In the remainder of this section, we optimize the three terms in Equation 8 sequentially. We now concentrate on the first term. For each class $c$, we minimize the class-conditional OT distance $W(\mu_{\text{true}}(\mathbf{x} \mid c), \nu_{\text{distill}}^{(\text{hard})}(\mathbf{x} \mid c))$ through diffusion guidance: we compute the OT distance between the distilled images and the real images in the latent space as the guiding function. At each diffusion step during the generation of the $n$-th latent $\mathbf{z}_0^c$, we draw a random batch of class-$c$ samples from dataset $\mathcal{T}$ and encode them into latent representations $\mathbf{Z}_{\mathcal{T}}^c$. We then employ the following guidance function:

$$\mathcal{G}_{\text{W}}(\mathbf{z}_t^c) = W([\mathbf{M}_{n-1}^c, \mathbf{z}_t^c], \mathbf{Z}_{\mathcal{T}}^c) \tag{9}$$

where $\mathbf{M}_{n-1}^c$ denotes previously sampled $n-1$ latents for class $c$, and $[\cdot]$ represents the pythonic concatenation. We denote $[\mathbf{M}_{n-1}^c, \mathbf{z}_t^c]$ as $\hat{\mathbf{M}}_n^c$. The OT matrix $\mathbf{P}^{\lambda_1}$ can be efficiently approximated:

$$\mathbf{P}^{\lambda_1} = \underset{\mathbf{P}}{\text{argmin}} \left\langle \mathbf{P}, \mathbf{D}(\hat{\mathbf{M}}_n^c, \mathbf{Z}_{\mathcal{T}}^c) \right\rangle - \lambda_1 h(\mathbf{P}), \text{ where } \sum_i \mathbf{P}_{ij} = \frac{1}{|\mathbf{Z}_{\mathcal{T}}^c|} \ \forall j, \ \sum_j \mathbf{P}_{ij} = \frac{1}{n} \ \forall i. \tag{10}$$

where $h(\mathbf{P})$ is the entropy of $\mathbf{P}$, $\langle \cdot, \cdot \rangle$ denotes the Frobenius inner product, $\lambda_1 > 0$ is the entropy regularization weight, $\mathbf{D}(\hat{\mathbf{M}}_n^c, \mathbf{Z}_{\mathcal{T}}^c)$ represents the cost matrix measuring the pairwise distance between the real latent representations $\mathbf{Z}_{\mathcal{T}}^c$ and the sampled $\hat{\mathbf{M}}_n^c$. Without loss of generality, we use the $\ell_p$-norm cost matrix and initialize the candidate transport matrix $\mathbf{K}^0$ as:

$$\mathbf{D}_{ij}(\hat{\mathbf{M}}_n^c, \mathbf{Z}_{\mathcal{T}}^c) = \| \hat{\mathbf{M}}_n^c[i] - \mathbf{Z}_{\mathcal{T}}^c[j] \|_p, \quad \mathbf{K}^0 = \exp(-\frac{\mathbf{D}}{\lambda_1}). \tag{11}$$

Next, Sinkhorn normalization is applied through iterative updates to $\mathbf{K}$:

$$\widehat{\mathbf{K}}^i \leftarrow \text{diag}\left(\mathbf{K}^{i-1}\mathbf{1}_n \oslash (n\mathbf{1}_{|\mathbf{Z}_{\mathcal{T}}^c|})\right)^{-1} \mathbf{K}^{i-1}, \ \ \mathbf{K}^i \leftarrow \widehat{\mathbf{K}}^i \text{diag}\left(\left(\widehat{\mathbf{K}}^i\right)^{\text{T}} \mathbf{1}_{|\mathbf{Z}_{\mathcal{T}}^c|} \oslash (|\mathbf{Z}_{\mathcal{T}}^c|\mathbf{1}_n))\right)^{-1}, \tag{12}$$

where $\oslash$ denotes element-wise division, $(\cdot)^{\text{T}}$ indicates matrix transpose. After $T$ iterations, the optimal transport matrix $\mathbf{P}^{\lambda_1}$ is obtained, and we can compute the Sinkhorn distance (an approximation of the OT distance) W as:

$$\mathbf{P}^{\lambda_1} = \mathbf{K}^T, \quad W([\mathbf{M}_{n-1}^c, \mathbf{z}_t^c], \mathbf{Z}_{\mathcal{T}}^c) = \left\langle \mathbf{P}, \mathbf{D}(\hat{\mathbf{M}}_n^c, \mathbf{Z}_{\mathcal{T}}^c) \right\rangle = \sum_{i,j} \mathbf{K}_{ij}^T \mathbf{D}_{ij} \tag{13}$$

For the entire guided diffusion sampling, we follow the previous approach to combine the terms of trajectory and diversity functions with our OT function. The iteration of $t = T_D \to 1$ yields $\mathbf{z}_0^c$:

$$\mathbf{z}_{t-1}^c = s(\mathbf{z}_t^c, t, \epsilon_\phi) - \rho_t \nabla_{\mathbf{z}_t^c} \mathcal{G}_I(\mathbf{z}_t^c, t) - \gamma_t \nabla_{\mathbf{z}_t^c} \mathcal{G}_D(\mathbf{z}_t^c) - \beta_1 \nabla_{\mathbf{z}_t^c} \mathcal{G}_W(\mathbf{z}_t^c), \tag{14}$$

where $\rho_t$, $\gamma_t$ and $\beta_1$ are weights, $\mathcal{G}_I(\mathbf{z}_t^c, t)$ and $\mathcal{G}_D(\mathbf{z}_t^c)$ are trajectory and diversity functions, respectively. By minimizing the OT distance in the image distribution space, we account for the contribution of individual real images and incorporate both global and local structural information, thereby promoting fine-grained geometric alignment between distributions. Finally, we use decoder $D$ to convert all latent representations into images, forming the distilled image set $\mathcal{S}_\mathbf{x}$.

## 4.4 Label-Image-Aligned Soft Label Relabeling (LIA)

We now focus on the contraction factor $\alpha(\nu_{\text{distill}}^{(\text{soft})})$, which characterizes the alignment between the complexity of the soft label distribution and that of the distilled image distribution (Appendix G.2 for details). Since the representational capacity of the distilled dataset is primarily governed by the number of images per class (IPC), we adopt an IPC-aware strategy that minimizes the overall OT distance. In low-IPC regimes, the distilled image distribution is less expressive and more prone to overfitting. Assigning overly complex soft labels in such cases can introduce distributional mismatch and degrade alignment. To mitigate this, we employ a smaller number of representative teacher models to produce simplified, low-entropy soft labels that offer well-calibrated supervision. In contrast, high-IPC regimes enable the distilled dataset to support greater semantic diversity. Accordingly, we leverage a larger and more diverse set of teacher models to generate fine-grained soft label distributions, which better capture the intrinsic structure of the true label space. Formally, for each synthetic image, we assign the soft label as the averaged output from a set of IPC-dependent teachers:

$$\mathbf{t}(\mathbf{x}_i) = \frac{1}{|\mathbb{T}(|\mathcal{S}_\mathbf{x}|)|} \sum_{t \in \mathbb{T}(|\mathcal{S}_\mathbf{x}|)} F_t(\mathbf{x}_i) = \frac{1}{|\mathbb{T}(\text{IPC})|} \sum_{t \in \mathbb{T}(\text{IPC})} F_t(\mathbf{x}_i), \quad \text{for each } \mathbf{x}_i \in \mathcal{S}_\mathbf{x}, \tag{15}$$

where $F_t$ denotes the logit output function of the $t$-th teacher, $\mathbf{t}(\mathbf{x}_i)$ denotes soft label for image $\mathbf{x}_i$, and $\mathbb{T}(\text{IPC})$ is a subset of teacher models selected to minimize the contraction factor $\alpha$. This strategic relabeling ensures that the soft label distribution faithfully matches the capacity of the distilled images, reducing the discrepancy term $\text{W}(\nu_{\text{distill}}^{(\text{soft})}, \mu_{\text{true}})$ and thereby improving alignment. For a fair comparison with prior methods [27, 28], we adopt the same region-level soft label storage strategy as in FKD [48].

## 4.5 OT-based Student Model Logit Matching (OTM)

After obtaining a distilled set that preserves rich geometric structures of the real data, we transfer this information to student models (i.e., new models) by aligning the distribution of their logits with that of the real dataset. We achieve this alignment by minimizing the last term in the upper bound of Equation 8. We consider a batch of $b$ samples and denote the soft labels of this batch and the logit output of a student model as $\mathbf{t}$ and $\mathbf{s}$, respectively. Most traditional divergence measures operate on a per-sample basis and match logits independently, thereby failing to capture inter-sample relationships. To address this limitation, we employ a batch-wise OT distance that aligns logits while capturing global distributional structure. Specifically, similar to Section 4.3, we use the Sinkhorn method to efficiently solve for the OT matrix $\mathbf{P}^{\lambda_2}$, with entropy regularization $h(\mathbf{P})$ weighted by $\lambda_2$:

$$\mathbf{P}^{\lambda_2} = \underset{\mathbf{P}}{\arg\min} \langle \mathbf{P}, \mathbf{C}(\mathbf{t}, \mathbf{s}) \rangle - \lambda_2 h(\mathbf{P}), \quad \text{where} \sum_i \mathbf{P}_{ij} = \frac{1}{b} \forall j, \sum_j \mathbf{P}_{ij} = \frac{1}{b} \forall i. \tag{16}$$

Here, $\mathbf{C} \in \mathbb{R}^{b \times b}$ is the batch-wise cost matrix, where each entry $\mathbf{C}_{ij}$ measures the distance between the soft label $\mathbf{t}(\mathbf{x}_i)$ and the synthetic output $\mathbf{s}(\mathbf{x}_j)$. Specifically, we employ the $\ell_p$-norm:

$$\mathbf{C}_{ij}(\mathbf{t}, \mathbf{s}) = \| \mathbf{t}(\mathbf{x}_i) - \mathbf{s}(\mathbf{x}_j) \|_p, \quad \mathcal{L}_{\text{SD}} = \text{W}(\mathbf{t}, \mathbf{s}) = \langle \mathbf{P}^{\lambda_2}, \mathbf{C} \rangle, \tag{17}$$

Adapting Equations 11, 12, and 13 to current dimensions, we compute $\mathbf{P}^{\lambda_2}$ and the corresponding batch-wise Sinkhorn distance loss $\mathcal{L}_{\text{SD}}$. For a fair comparison with previous methods [40, 23], we use the cross-entropy loss $\mathcal{L}_{\text{CE}}$, the MSE loss $\mathcal{L}_{\text{MSE}}$, and the Sinkhorn loss $\mathcal{L}_{\text{SD}}$ for distillation:

$$\mathcal{L} = \sum_{i=1}^b \kappa_1 \mathcal{L}_{\text{CE}}(y_{\text{onehot}}(\mathbf{x}_i), \mathbf{s}(\mathbf{x}_i)) + \kappa_2 \mathcal{L}_{\text{MSE}}(\mathbf{t}(\mathbf{x}_i), \mathbf{s}(\mathbf{x}_i)) + \beta_2 \mathcal{L}_{\text{SD}}, \tag{18}$$

where $\kappa_1$, $\kappa_2$, and $\beta_2$ are scalar weights, $y_{\text{onehot}}(\mathbf{x}_i)$ denotes the hard label for the distilled image $\mathbf{x}_i$.

Table 2: Performance comparison on ImageNet-1K [22] with ResNet-18. The numbers in parentheses for "Ours" represent the number of training epochs on the distilled set for new models.

| | | | | Comparsion with generative-model-based methods. | | | | | |
|---|---|---|---|---|---|---|---|---|---|
| IPC | $D^3M$ [31] | $D^4M$ [32] | TDSDM [33] | DiT [44] | Minimax [45] | DDPS [46] | DiT-IGD [24] | Ours (300) | Ours (1000) |
| 10 | 23.6±0.1 | 27.9±0.7 | 44.5±0.4 | 39.6±0.4 | 42.1±0.3 | 42.1±0.3 | 45.5±0.5 | **52.9±0.1** | **58.6±0.3** |
| 50 | 32.2±0.1 | 55.2±0.3 | 59.4±0.3 | 52.9±0.6 | 59.4±0.2 | 59.4±0.2 | 59.8±0.3 | **61.9±0.5** | **64.2±0.4** |

| | | | | Comparsion with model-inversion-based methods | | | | | |
|---|---|---|---|---|---|---|---|---|---|
| IPC | $SRe^2L$ [27] | G-VBSM [40] | RDED [28] | CDA [26] | SC-DD [41] | EDC [23] | CV-DD [25] | Ours (300) | Ours (1000) |
| 10 | 21.3±0.6 | 31.4±0.5 | 42.0±0.1 | 33.5±0.3 | 32.1±0.2 | 48.6±0.3 | 46.0±0.6 | **52.9±0.1** | **58.6±0.3** |
| 50 | 46.8±0.2 | 51.8±0.4 | 56.5±0.1 | 53.5±0.3 | 53.1±0.1 | 58.0±0.2 | 49.5±0.4 | **61.9±0.5** | **64.2±0.4** |

Table 3: Other architecture performance comparison on ImageNet-1K [22].

| Method | MobileNet-V2 | | EfficientNet-B0 | | Swin Transformer | | ConvNeXt | |
|---|---|---|---|---|---|---|---|---|
| | IPC10 | IPC50 | IPC10 | IPC50 | IPC10 | IPC50 | IPC10 | IPC50 |
| $SRe^2L$ [27] | 10.2±2.6 | 31.8±0.3 | 11.4±2.5 | 34.8±0.4 | 4.8±0.6 | 42.1±0.3 | 4.1±0.4 | 48.8±0.2 |
| RDED [28] | 40.4±0.1 | 53.3±0.2 | 31.0±0.1 | 58.5±0.4 | 42.3±0.6 | 53.2±0.8 | 48.3±0.5 | 65.4±0.4 |
| EDC [23] | 45.0±0.2 | 57.8±0.1 | 51.1±0.3 | 60.9±0.2 | 46.0±0.5 | 57.9±0.3 | 54.4±0.2 | 66.6±0.2 |
| DiT-IGD [44] | 39.2±0.2 | 57.8±0.2 | 47.7±0.1 | 62.0±0.1 | 44.1±0.6 | 58.6±0.5 | 51.9±0.2 | 66.8±0.5 |
| Ours (300) | **51.0±0.6** | **61.0±0.4** | **56.7±0.2** | **64.4±0.1** | **50.2±0.2** | **68.2±0.1** | **61.2±0.1** | **70.2±0.8** |
| Ours (500) | **54.6±0.3** | **63.0±0.4** | **59.6±0.2** | **66.0±0.6** | **56.2±1.0** | **69.4±0.1** | **64.5±0.3** | **71.1±1.1** |
| Ours (1000) | **57.6±0.1** | **63.9±0.2** | **62.4±0.1** | **66.8±0.1** | **63.7±0.2** | **70.5±0.1** | **67.0±0.1** | **71.8±0.9** |

# 5 Experiments

## 5.1 Experimental Settings

**Dataset.** Given our primary focus on large-scale dataset distillation, we evaluate our method on the full ImageNet-1K dataset [22]. To ensure comparability across varying category scales, we further conduct experiments on two widely used subsets, ImageNet-100 [13] and ImageNette [49]. To construct a comprehensive benchmark covering both low-resolution and high-resolution settings, we additionally include CIFAR-100 [21]. The dataset descriptions are presented in Appendix C.

**Network architectures.** To evaluate the generalization capability of our method, we experiment with a diverse set of network architectures, including convolutional neural networks (CNNs), transformer-based models, and hybrid models. Specifically, we consider CNN-based architectures, including ResNet [50], MobileNet [51],EfficientNet [52], and ConvNet [53]; a transformer-based model, the Swin Transformer [54]; and the hybrid architecture ConvNeXt [55]. This selection provides a comprehensive evaluation across diverse architectural paradigms and inductive biases.

**Baselines.** We compare our approach with a broad range of dataset distillation methods. Specifically, we include traditional methods such as DM [16], IDC [13], and DATM [56]; model-inversion-based methods including $SRe^2L$ [27], G-VBSM [40], RDED [28], CDA [26], SC-DD [41], EDC [23], CV-DD [25], and DELT[57]; as well as generative-model-based methods such as $D^3M$ [31], $D^4M$ [32], TDSDM [33], DiT [44], Minimax [45], DDPS [46], and IGD [24]. We report the top-1 test accuracy of models trained on distilled datasets with different IPC (Images Per Class) settings to ensure a fair and consistent comparison. Each network is trained five times from scratch to report error bars.

**Implementation details.** To ensure fair evaluation, we follow the configurations of IGD [24] and EDC [23], maintaining consistency in training procedure and hyperparameter settings. For the OT components, we set $\alpha_1 = 1$, $\gamma_1 \in \{1000, 3000\}$, $\alpha_2 = 0.1$, and $\gamma_2 = 0.1$. For simplicity, we set $p = 1$ ($\ell_1$-norm). More details are in Appendix F.

## 5.2 Results and Discussions

**Results on ImageNet-1K.** We extensively evaluated our generative OT framework on ImageNet-1K [22], comparing it against state-of-the-art dataset distillation methods, including both generative model-based and model-inversion-based approaches, across various architectures and IPC settings. Table 2 presents results on ResNet-18 [50]. Our method significantly outperforms prior methods

Table 4: Performance comparison on ImageNette [49].

| Model | ConvNet-6 | | | ResNetAP-10 | | | ResNet-18 | | |
|---|---|---|---|---|---|---|---|---|---|
| IPC | 10 | 50 | 100 | 10 | 50 | 100 | 10 | 50 | 100 |
| Hard Label | | | | | | | | | |
| Random | 46.0±0.5 | 71.8±1.2 | 79.9±0.8 | 54.2±1.2 | 77.3±1.0 | 81.1±0.6 | 55.8±1.0 | 75.8±1.1 | 82.0±0.4 |
| DM [16] | 49.8±1.1 | 70.3±0.8 | 78.5±0.8 | 60.2±0.7 | 76.7±1.1 | 80.9±0.7 | 60.9±0.7 | 75.0±1.0 | 81.5±0.4 |
| IDC-1 [13] | 48.2±1.2 | 72.4±0.7 | 80.6±1.1 | 60.4±0.6 | 77.4±0.7 | 81.5±1.2 | 61.0±0.8 | 77.5±1.0 | 81.7±0.8 |
| DiT [44] | 56.2±1.3 | 74.1±0.6 | 78.2±0.3 | 62.8±0.8 | 76.9±0.5 | 80.1±1.1 | 62.5±0.9 | 75.2±0.7 | 77.8±0.7 |
| Minimax [45] | 58.2±0.9 | 76.9±0.8 | 81.1±0.3 | 63.2±1.0 | 78.2±0.7 | 81.5±1.0 | 64.9±0.6 | 78.1±0.6 | 81.3±0.7 |
| DiT-IGD [44] | 61.9±1.9 | 80.9±0.9 | 84.5±0.7 | 66.5±1.1 | 81.0±1.2 | 85.2±0.8 | 67.7±0.3 | 80.4±0.8 | 84.4±0.8 |
| Ours | **67.0±0.9** | **83.1±1.0** | **86.5±0.5** | **68.0±0.3** | **83.8±0.6** | **86.4±0.6** | **69.1±1.9** | **84.6±0.4** | **85.9±0.2** |
| Soft Label | | | | | | | | | |
| SRe$^2$L [27] | - | - | - | - | - | - | 29.4±3.0 | 40.9±0.3 | 50.2±0.4 |
| RDED [28] | 63.5±0.6 | 84.3±0.3 | 89.2±0.7 | 60.8±0.5 | 80.5±0.3 | 89.3±0.6 | 61.4±0.4 | 80.4±0.4 | 89.6±1.0 |
| D$^4$M [32] | 53.5±0.5 | 84.4±0.4 | 89.6±0.2 | 56.2±0.3 | 84.7±0.5 | 90.2±0.3 | 57.4±0.4 | 84.8±0.2 | 90.4±0.7 |
| DDPS$^c$ [46] | - | - | - | - | - | - | 62.5±0.2 | 83.4±0.5 | 90.2±0.2 |
| DDPS$^s$ [46] | - | - | - | - | - | - | 60.4±0.3 | 85.8±0.4 | 91.6±0.4 |
| DiT-IGD* [24] | 69.6±1.0 | 86.7±0.9 | 89.9±0.6 | 73.6±1.3 | 86.8±1.0 | 90.6±0.6 | 74.8±0.7 | 86.4±0.9 | 90.7±0.5 |
| Ours | **74.5±0.3** | **89.1±0.9** | **91.3±0.2** | **77.8±0.8** | **89.7±0.5** | **91.6±0.3** | **79.0±0.3** | **89.3±0.3** | **92.0±0.6** |
| Full | 94.3±0.5 | | | 94.6±0.5 | | | 95.3±0.6 | | |

Table 5: Performance comparison on ImageNet-100 [13].

| Model | IPC | SRe$^2$L [27] | RDED [28] | DELT [57] | Ours |
|---|---|---|---|---|---|
| ResNet-18 | 10 | 9.5±0.4 | 36.0±0.3 | 28.2±1.5 | **47.7±0.3** |
| | 50 | 27.0±0.4 | 61.6±0.1 | 67.9±0.6 | **72.6±0.1** |
| | 100 | 30.4±0.3 | 74.5±0.4 | 75.1±0.2 | **79.2±0.1** |
| ResNet-101 | 10 | 6.4±0.1 | 33.9±0.1 | 22.4±3.3 | **36.3±0.5** |
| | 50 | 25.7±0.3 | 66.0±0.6 | 70.8±2.3 | **74.3±0.2** |
| | 100 | 27.6±0.2 | 73.5±0.8 | 77.6±1.8 | **81.6±0.1** |
| MobileNet | 10 | 4.5±0.4 | 23.6±0.7 | 15.8±0.2 | **43.2±0.2** |
| | 50 | 18.4±0.2 | 51.5±0.8 | 55.0±1.8 | **69.5±0.3** |
| | 100 | 22.1±0.3 | 70.8±1.1 | 76.7±0.3 | **78.0±0.2** |

Table 6: Performance comparison on CIFAR-100 [21] using ConvNet-3 [53].

| IPC | 10 | 50 | 100 |
|---|---|---|---|
| DM [16] | 29.7±0.3 | 43.6±0.4 | 47.1±0.4 |
| M3D [18] | 42.4±0.2 | 50.9±0.7 | 52.1±0.6 |
| DATM [56] | 47.2±0.4 | 55.0±0.2 | 57.5±0.2 |
| SRe$^2$L [27] | 24.5±0.4 | 45.2±0.3 | 46.6±0.5 |
| RDED [28] | 48.1±0.3 | 57.0±0.1 | 58.1±0.4 |
| D$^4$M [32] | 45.0±0.1 | 48.8±0.3 | 50.3±0.2 |
| DiT-IDG [24] | 45.8±0.6 | 53.9±0.6 | 55.9±0.4 |
| Ours | **50.7±0.2** | **57.5±0.3** | **58.7±0.2** |

at 300 epochs. When training is extended to 1000 epochs, performance further improves. This shows that our distilled images and soft labels contain sufficient information for continued optimization. Beyond ResNet, we evaluated generalization on MobileNet-V2 [51], EfficientNet-B0 [52], Swin Transformer [54], and ConvNeXt [55] (Table 3). Our framework consistently surpasses prior approaches across all architectures. The larger performance gain at lower IPC settings highlights its ability to better preserve fine-grained distributional details. When IPC is low, existing dataset distillation methods struggle to cover the full data distribution, leading to significant discrepancies between the learned distribution and the real distribution. In contrast, our approach explicitly aligns the latent space distribution, logit-level semantic consistency, and label-image relationships, ensuring that even with limited synthetic samples, our distilled set comprehensively represents the real data.

**Results on ImageNet subsets.** To further compare with prior works and to evaluate our method under reduced-category settings, we conduct experiments on two ImageNet subsets, varying both class selection and class count. As shown in Tables 4 and 5, our method consistently outperforms all baselines. Notably, we observe significant performance improvements under both hard label and soft label settings. This demonstrates that OT-guided sampling effectively captures fine-grained sample information, contributing to the learning of the new model. During the subsequent OT distance minimization phases, this extracted information is systematically transferred to the new model, resulting in enhanced performance. Robustness tests are conducted in Appendix G.8.

**Results on CIFAR-100.** We evaluate our method on CIFAR-100 [21] to assess its generalizability on low-resolution datasets, as summarized in Table 6. To ensure a broad comparison, we include traditional low-resolution-oriented methods, along with model-inversion-based and generative-model-based methods, both specially designed for large-scale datasets. Unlike most existing methods that specialize in either low-resolution or high-resolution datasets, our approach achieves state-of-the-art performance on ImageNet [22] while maintaining superior results on CIFAR. This further highlights the robustness of our OT-driven strategy in preserving distributional characteristics across scales.

Table 7: Ablation Study on ImageNette [49] under IPC=10. *Note*: Here, "w/o LIA" denotes soft relabeling with the teacher ensemble from high-IPC settings, without adapting to the current IPC.

| Model | Hard Label | | Soft Label | | | |
|---|---|---|---|---|---|---|
| | w/o OTG | w OTG | w/o OTG | w/o LIA | w/o OTM | Full |
| ConvNet-6 | 61.9 | **67.0** | 72.5 | 74.3 | 73.2 | **74.5** |
| ResNetAP-10 | 66.5 | **68.0** | 74.2 | 76.4 | 75.9 | **77.8** |
| ResNet-18 | 67.7 | **69.1** | 77.2 | 77.8 | 77.5 | **79.0** |

Table 8: Mean runtime per class (sampling) or per epoch (matching) on ImageNet-1K [22].

| Stage | Method | IPC=10 | IPC=50 |
|---|---|---|---|
| Samp. | w/o OTG | 97.1s | 537.4s |
| | w OTG | 97.7s | 540.3s |
| Match. | w/o OTM | 23.2s | 126.1s |
| | w OTM | 23.3s | 126.6s |

Table 9: Distilled set generation time (IPC=10, ImageNet-1K, 8×4090). PreS: Presample, PostS: Postsample.

| EDC [23] | 3h PreS +3h PostS + 5h Recover + 0.4h Relabel |
|---|---|
| Ours | 3.4h Diffusion Sample + 0.3h Relabel |

Table 10: Effect of $\alpha$ on ImageNette [49].

| Teachers | ResNet-18 | w/o LIA | w LIA |
|---|---|---|---|
| $\alpha$ | 0.906 | 0.903 | **0.643** |
| Avg. Acc. | 76.0 | 76.2 | **77.1** |

**Impact of different components.** Our OT-guided diffusion sampling effectively transfers the geometric structure of the image space distribution to the distilled images. This alignment is further enhanced by the Label-Image-Aligned Soft Relabeling, which narrows the distributional gap between the distilled and real data. During student model training, the OT-based student logit matching module faithfully propagates this information to the new model. This further reinforces alignment between the original distribution $\mu_{\text{true}}$ and the learned distribution $\nu_{\text{new}}$. As shown in Table 7, each component involved in minimizing the OT distance plays a critical role, underscoring the necessity of aligning distributions throughout the entire pipeline. More validations are provided in Appendix G.3.

**Runtime analysis.** As shown in Table 8, the additional time overhead introduced by our OT constraint is consistently less than 1%. Table 9 provides a breakdown of the time required for each step in generating the distilled set for both our method and the state-of-the-art model-inversion-based method, EDC [23]. Our approach is notably faster than EDC, which further demonstrates its efficiency.

**Discussion of contraction factor $\alpha$.** Table 10 reports the values of $\alpha$ measured under different soft label generation strategies. Our LIA strategy significantly reduces the OT distance between the distilled data and the real data, allowing the distilled data to capture more information of the real distribution. This leads to a substantial performance improvement. More discussion in Appendix G.2.

**Sensitivity analysis.** As shown in Table 11, our method delivers consistently high accuracy over a broad range of OT-related hyperparameter settings, demonstrating low sensitivity. This robustness eliminates exhaustive tuning and enables straightforward deployment across diverse scenarios. It also makes our approach readily scalable. Additional results and analyses are available in Appendix G.1.

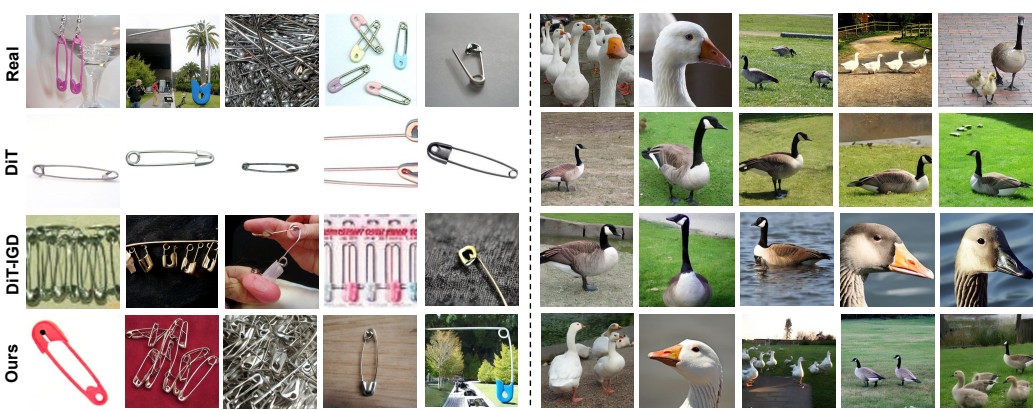

Figure 2: Comparison of generated images from different methods on ImageNet-100 (IPC = 10).

Table 11: Impact of different OT hyperparameters.

| $\beta_1$ | $\lambda_1$ | $\beta_2$ | $\lambda_2$ | ConvNet | ResNetAP | ResNet |
|---|---|---|---|---|---|---|
| | | | | Hard Label | | |
| 1 | 1000 | - | - | 67.0±0.9 | 68.0±0.3 | 69.1±1.9 |
| 1 | 10000 | - | - | 66.3±0.7 | 68.5±0.3 | 68.9±0.8 |
| 10 | 1000 | - | - | 65.8±0.5 | 67.5±0.5 | 68.7±1.1 |
| | | | | Soft Label | | |
| 1 | 1000 | 0.1 | 0.1 | 74.5±0.3 | 77.8±0.8 | 79.0±0.3 |
| 1 | 1000 | 0.1 | 1 | 74.6±0.8 | 76.3±0.8 | 78.2±1.0 |
| 1 | 1000 | 1 | 0.1 | 74.3±0.5 | 78.1±0.3 | 77.4±0.2 |

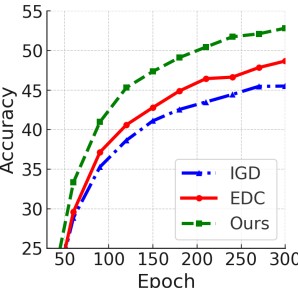 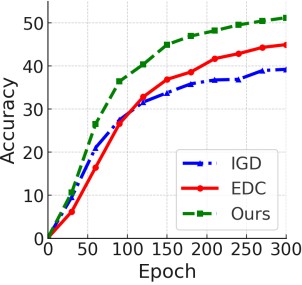

Figure 3: Training accuracy curves of ResNet-18 (left) and MobileNet (right). IGD: DiT-IGD.

**Visualizations.** Figure 2 illustrates a qualitative comparison among DiT [44], DiT-IGD [24], and our method. DiT often produces visually similar outputs that lack semantic diversity. DiT-IGD introduces diversity without aligning with the underlying real data distribution, leading to non-representative or incorrect generations. Furthermore, its influence estimation is based solely on intraclass averaged statistics, which results in perceptible blurring. In contrast, our approach explicitly models both instance-specific characteristics and fine-grained distributional structures, thereby enabling faithful approximation of the real data manifold. We also present the test accuracy at each logit-matching step in Figure 3. Our method achieves faster convergence and consistently higher accuracy, especially in early epochs, demonstrating superior sample informativeness and stronger distribution alignment when compared to EDC [23] and DiT-IGD [24]. Please refer to Appendix H for more visualizations.

# 6 Conclusion

We propose a principled framework for generative large-scale dataset distillation by formulating it as an OT distance minimization problem. Our approach explicitly decomposes the total OT distance into three interpretable components and systematically minimizes each to ensure comprehensive distributional alignment. This allows new models trained on distilled data to behave similarly to models trained on the full dataset, regardless of architecture. Extensive experiments across diverse datasets and model architectures validate the effectiveness and generalizability of our method.

**Broader Impact.** Our distilled datasets lower carbon footprints associated with new model training, fostering sustainable AI development. They also enable efficient learning in federated and continual learning scenarios, enhancing data privacy and model adaptation across distributed systems.

**Acknowledgement** This work was supported by the GPU cluster built by MCC Lab of Information Science and Technology Institution, USTC, and the Supercomputing Center of the USTC.

**Competing Interests** The authors declare no competing interests.

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

# Appendix

## A   Overview

This appendix provides comprehensive supplementary materials to further elaborate on our method's theoretical foundations, experimental setup, and empirical findings. It includes the following sections:

- **Section B: More Related Work.** Detailed discussions on prior studies, with an emphasis on optimal transport and guided diffusion sampling

- **Section C: Dataset Descriptions.** Comprehensive descriptions of all datasets used in our experiments, including ImageNet-1K [22], ImageNette [49], ImageNet-100 [13], and CIFAR-100 [21].

- **Section D: Symbol Table.** A complete summary of key mathematical notations, hyperparameters, and definitions referenced throughout the paper.

- **Section E: Pseudocode.** Step-by-step pseudocode for the proposed pipeline, detailed procedures for OT-alignment in different stages, and the calculation process for the contraction factor $\alpha$.

- **Section F: Implementation Details.** Full specifications of hyperparameter settings, training schedules, and augmentation strategies across all datasets used in our experiments.

- **Section G: Further Experimental Analyses.** Additional experiments, including sensitivity studies (G.1), in-depth analysis of the contraction factor (G.2), comparisons with alternative distance metrics (G.3), runtime analysis (G.4), data coverage evaluation (G.5), expanded comparisons with other baselines (G.6 and G.7), robustness evaluation under adversarial attacks (G.8) and evaluation under extremely low-IPC settings G.9.
  *This section constitutes the core of the appendix, offering deep empirical analyses of the contraction factor $\alpha$ and the robustness properties of the distilled models. Other experiments further strengthen and extend the key findings presented in the main text.*

- **Section H: More Visualization Results.** Additional qualitative visualizations, including t-SNE plots and synthesized images, to assess semantic coverage and distributional diversity.

- **Section I: Limitations.** Critical discussion of the imitations of our framework.

- **Section J: Broader Impact.** Reflections on the broader societal, ethical, and practical implications of our dataset distillation method.

Together, these supplementary materials provide a complete and transparent view of our method, support full reproducibility, and offer additional insights that complement and strengthen the main paper.

## B   More Related Work

**Optimal transport**   OT theory provides a principled mathematical framework for comparing probability distributions by computing the minimal cost required to transform one distribution into another. Compared to KL divergence and Jensen-Shannon (JS) divergence, OT provides a more geometrically faithful measure of distributional differences, particularly when dealing with distributions with non-overlapping supports [58, 59]. The Wasserstein distance, also known as OT distance, effectively quantifies distributional discrepancies and has been widely applied in image generation [60, 61, 62], causal discovery [63, 64], unsupervised learning [65, 66, 67, 68], and reinforcement learning [69, 70, 71, 72]. However, its exact computation is intractable for high-dimensional data due to prohibitive complexity. To overcome this, the Sinkhorn distance introduces entropy regularization, making OT computation more efficient and numerically stable [73]. This regularized variant extends OT applications to domain adaptation [74, 75, 76], classification [77, 78], and knowledge distillation [34, 79, 80]. In this work, we propose a generative model-based OT framework designed to achieve precise distributional alignment throughout the dataset distillation process. Our approach optimizes the distilled dataset to minimize the OT distance between any student model's output distribution and the real data distribution, ensuring improved generalization.

**Guided diffusion sampling** Guided diffusion sampling enhances the generative capabilities of pre-trained diffusion models by incorporating external guidance during the reverse process to steer generation toward desired semantics [81]. Early methods, such as classifier guidance [82], inject gradients from pre-trained classifiers into the sampling process to condition generation. However, this approach necessitates domain-specific classifiers trained on noisy intermediate latents, which are often impractical. Classifier-free guidance [83] addresses this limitation by training the model with both conditional and unconditional objectives, enabling control without external models. Building upon this, Wang et al. [84] introduced linear operator-based guidance, constraining the diffusion process to the null space of known measurement operators; nonetheless, this strategy faces challenges in generalizing to nonlinear mappings. Subsequent works [85, 86, 87] extend guidance to inverse problems through iterative optimization and plug-and-play conditioning. Concurrently, methods like those proposed by Gopalakrishnan Nair et al. [88], Yu et al. [47], and Bansal et al. [89] introduce generic guidance functions by injecting gradients from task-specific losses computed on denoised intermediate states, thereby broadening applicability without necessitating model retraining. Inspired by these methods, Influence-Guided Diffusion (IGD) [24] leverages guided diffusion for dataset distillation by modifying the reverse sampling process to generate training-optimal data. However, its reliance on matching global distributional trajectories and introducing diversity through random perturbations often leads to suboptimal alignment, neglecting discriminative yet informative local characteristics in favor of global averaging. To overcome this limitation, we propose an optimal transport-based guidance strategy that explicitly aligns the geometric structure of real and synthetic distributions, achieving fine-grained consistency in guided diffusion sampling.

## C  Dataset Description

**ImageNet-1K** ImageNet-1K [22], also known as the ILSVRC 2012 dataset, is a large-scale image classification benchmark comprising 1,000 object categories. It contains approximately 1.28 million training images, 50,000 validation images, and 100,000 test images. The dataset is organized according to the WordNet hierarchy, with each synset corresponding to a distinct semantic concept. ImageNet-1K has been instrumental in advancing deep learning research and remains a standard benchmark for evaluating image classification models in large-scale settings.

**ImageNette** ImageNette [49] is a curated subset of ImageNet, consisting of 10 relatively easy categories, including "tench", "English springer", "cassette player", "chain saw", "church", "French horn", "garbage truck", "gas pump", "golf ball", and "parachute". It was introduced to facilitate rapid experimentation and prototyping of image classification models, particularly under limited computational budgets. All images are resized to a resolution of $224 \times 224$ pixels, providing a lightweight yet meaningful benchmark for distillation and robustness studies.

**ImageNet-100** ImageNet-100 [13] is another subset derived from ImageNet-1K, comprising 100 randomly selected classes. Each class typically contains around 1,000 training images and 300 test images, maintaining a relatively balanced distribution. ImageNet-100 provides a manageable yet challenging benchmark for evaluating classification performance, especially in scenarios where computational efficiency and rapid iteration are prioritized.

**CIFAR-100** CIFAR-100 [21] is a widely used benchmark dataset for image classification, extending the number of classes from 10 (CIFAR-10) to 100. Each class contains 600 images, with all images having a resolution of $32 \times 32$ pixels. Despite its compact size, CIFAR-100 presents a significant classification challenge due to its high intra-class variability and fine-grained label structure, making it a valuable resource for developing and assessing lightweight classification models.

## D  Symbol Description

To enhance clarity, a detailed description of mathematic symbols in the present study is provided in Table 12.

Table 12: Descriptions of all symbols, functions, and hyperparameters introduced in the main paper.

| Symbol | Definition |
|---|---|
| $E$ | Encoder to transform image into the latent space |
| $D$ | Decoder to reconstruct latent code back to the image space |
| $\mathcal{S}$ | Distilled (synthetic) dataset |
| $\mathcal{T}$ | Real (full) dataset |
| $\mathbf{x}$ | Image |
| $\mathbf{z}_0$ | Latent code of clean sample |
| $\mathbf{z}_t$ | Latent code of noisy sample at time step $t$ |
| $\mathbf{z}_t^c$ | Latent code of class-$c$ noisy sample at time step $t$ |
| $\alpha_t$ | Noise schedule controlling the perturbation at time step $t$ |
| $\boldsymbol{\epsilon}$ | Gaussian noise |
| $\epsilon_\phi$ | Denoising function parameterized by $\phi$ |
| $s$ | Reverse diffusion update function |
| $\mathcal{G}$ | Guidance function in guided diffusion |
| $\mathcal{G}_I$ | Influence function for general alignment |
| $\mathcal{G}_D$ | Diversity function enforcing diversity in distilled data |
| $\mathcal{G}_{\mathrm{W}}$ | Guidance function based on optimal transport |
| $\mathbf{M}_n^c$ | Previously sampled $n$ latents for class $c$ |
| $\hat{\mathbf{M}}_n^c$ | Concatenation of $\mathbf{M}_{n-1}^c$ and the latent $\mathbf{z}_t^c$ under sampling |
| $\mathbf{Z}_{\mathcal{T}}^c$ | Latent representations of class $c$ from the real dataset |
| $p$ | Norm order |
| $\mathbf{P}^{\lambda_1}$ | Optimal transport matrix for guided diffusion sampling with regularization |
| $\mathbf{D}$ | Latent space cost matrix for guided diffusion sampling |
| $\mathbf{K}^t$ | Transport matrix at the $t$-th step of Sinkhorn normalization |
| $T$ | Sinkhorn iterations |
| $T_D$ | Diffusion denoising iterations |
| $\mathbf{P}^{\lambda_2}$ | Optimal transport matrix for logit matching with regularization |
| $\mathbf{C}$ | Batch-wise cost matrix for logit matching |
| $F_t$ | The logit output function of the $t$-th teacher |
| $\mathbf{t}$ | Soft label for a batch |
| $\mathbf{s}$ | Student model output for a batch |
| $\mathbf{t}_i$ | Soft label for the $i$-th image in a batch |
| $\mathbf{y}_{\mathrm{onehot}}(\mathbf{x}_i)$ | One-hot hard label for the $i$-th image in a batch |
| $b$ | Batch size |
| $h(\mathbf{P})$ | The entropy of $\mathbf{P}$ |
| $\nu_{\mathrm{distill}}$ | Distilled data distribution |
| $\nu_{\mathrm{distill}}^{\mathrm{soft}}$ | Distilled data distribution with soft label |
| $\nu_{\mathrm{distill}}^{\mathrm{hard}}$ | Distilled data distribution with hard label |
| $\mu_{\mathrm{true}}$ | Real dataset distribution |
| $\nu_{\mathrm{new}}$ | Output of the student model after training on the distilled set |
| $\mathrm{W}(\mu_{\mathrm{true}}, \nu_{\mathrm{new}})$ | Wasserstein distance between real dataset and student model output |
| $\rho_t$ | Weight for influence function in the reverse sampling process |
| $\gamma_t$ | Weight for diversity function in the reverse sampling process |
| $\beta_1$ | Weight for the optimal transport guidance in the reverse sampling process |
| $\lambda_1$ | Entropy regularization weight for optimal transport matrix |
| $\lambda_2$ | Entropy regularization weight for logit matching |
| $\alpha(\nu_{\mathrm{distill}}^{(\mathrm{soft})})$ | Contraction factor quantifying the benefit of soft labels |
| IPC | Images per class in the dataset |
| $\mathbb{T}$ | Set of teacher models |
| $\mathcal{S}_{\mathbf{x}}$ | Distilled image set |
| $\kappa_1$ | Weight for cross-entropy loss in logit matching |
| $\kappa_2$ | Weight for mean squared error loss in logit matching |
| $\beta_2$ | Weight for Sinkhorn distance loss in logit matching |
| $\mathcal{L}_{\mathrm{CE}}$ | Cross-entropy loss |
| $\mathcal{L}_{\mathrm{MSE}}$ | Mean squared error loss |
| $\mathcal{L}_{\mathrm{SD}}$ | Sinkhorn distance loss |

# E PseudoCode

We present the pseudocode for our pipeline in Algorithm 1. The detailed calculation of the optimal transport (OT) distance for OT-guided Diffusion Sampling is provided in Algorithm 2, while the OT-based Student Model Logit matching is outlined in Algorithm 3. For efficient computation, we approximate the contraction factors using features in the latent space, enabling dimensionality reduction while preserving critical information.

---

**Algorithm 1** OT-based Generative Dataset Distillation Framework

---

**Require:** Real dataset $\mathcal{T} = \{(\mathbf{x}_i, y_i)\}$, teacher models $\mathbb{T}$, target IPC, diffusion model $G$, encoder $E$, decoder $D$, student model $S$
**Ensure:** Distilled dataset $\mathcal{S}_{\mathbf{x}}$ and trained student model $S$
 1: **for** each class $c = 1$ to $C$ **do**
 2:     Encode real samples: $\mathbf{Z}_{\mathcal{T}}^c \leftarrow E(\{\mathbf{x}_i : y_i = c\})$
 3:     **for** sample index $n = 1$ to IPC **do**
 4:         Sample latent $\mathbf{z}_{T_D}^c$ using diffusion model $G$
 5:         **for** $t = T_D$ to $1$ **do**
 6:             Compute OT-guidance $\mathcal{G}_{\mathrm{W}}(\mathbf{z}_t^c)$ w.r.t. $\mathbf{Z}_{\mathcal{T}}^c$ and previously sampled latents $\mathbf{M}_{n-1}^c$
 7:             Update latent using guidance: $\mathbf{z}_{t-1}^c \leftarrow s(\mathbf{z}_t^c, t, \epsilon_\phi) - \rho_t \nabla \mathcal{G}_I - \gamma_t \nabla \mathcal{G}_D - \beta_1 \nabla \mathcal{G}_{\mathrm{W}}$
 8:         **end for**
 9:         Append $\mathbf{z}_{t-1}^c$ to $\mathbf{M}^c$
10:     **end for**
11: **end for**
12: Decode all latents: $\mathcal{S}_{\mathbf{x}} \leftarrow D(\mathbf{M}_{\mathrm{IPC}}^c)$ for all $c$
13: Select teacher set $\mathbb{T}(\mathrm{IPC})$ according to IPC level
14: **for** each image $\mathbf{x}_i \in \mathcal{S}_{\mathbf{x}}$ **do**
15:     Generate soft label: $\mathbf{t}(\mathbf{x}_i) \leftarrow \frac{1}{|\mathbb{T}|} \sum_{t \in \mathbb{T}} F_t(\mathbf{x}_i)$
16: **end for**
17: **for** each training batch $\mathcal{B} \subset \mathcal{S}_{\mathbf{x}}$ **do**
18:     Get soft labels $\mathbf{t}$ and student outputs $\mathbf{s} \leftarrow S(\mathcal{B})$
19:     Compute batch-wise OT loss $\mathcal{L}_{\mathrm{SD}} \leftarrow \mathrm{W}(\mathbf{t}, \mathbf{s})$
20:     Compute per-sample CE and MSE loss: $\mathcal{L}_{\mathrm{CE}} = \sum \mathcal{L}_{\mathrm{CE}}(y_{\mathrm{onehot}}, \mathbf{s}), \quad \mathcal{L}_{\mathrm{MSE}} = \sum \mathcal{L}_{\mathrm{MSE}}(\mathbf{t}, \mathbf{s})$
21:     Total loss: $\mathcal{L} = \kappa_1 \mathcal{L}_{\mathrm{CE}} + \kappa_2 \mathcal{L}_{\mathrm{MSE}} + \beta_2 \mathcal{L}_{\mathrm{SD}}$
22:     Update student model $S$ using gradient descent
23: **end for**
24: **return** $\mathcal{S}_{\mathbf{x}}, S$

---

**Algorithm 2** Computation of OT-based Guidance for Image Latent Sampling

---

**Require:** Previously sampled latents $\mathbf{M}_{n-1}^c$, current latent $\mathbf{z}_t^c$, a random batch of real class latents $\mathbf{Z}_{\mathcal{T}}^c$, regularization weight $\lambda_1$, iteration number $T$
**Ensure:** Optimal transport distance as guidance value $\mathcal{G}_{\mathrm{W}}(\mathbf{z}_t^c)$
 1: Concatenate latent: $\hat{\mathbf{M}}_n^c \leftarrow [\mathbf{M}_{n-1}^c, \mathbf{z}_t^c]$
 2: Compute cost matrix: $\mathbf{D}_{ij} \leftarrow \|\hat{\mathbf{M}}_n^c(i) - \mathbf{Z}_{\mathcal{T}}^c(j)\|_p$
 3: Initialize kernel matrix: $\mathbf{K} \leftarrow \exp(-\mathbf{D}/\lambda_1)$
 4: Set iteration counter $t \leftarrow 0$
 5: **while** $t < T$ **do**
 6:     Row normalization: $\mathbf{K} \leftarrow \mathrm{diag}\left(\mathbf{K}\mathbf{1}_n \oslash (n \cdot \mathbf{1}_{|\mathbf{Z}_{\mathcal{T}}^c|})\right)^{-1} \cdot \mathbf{K}$
 7:     Column normalization: $\mathbf{K} \leftarrow \mathbf{K} \cdot \mathrm{diag}\left(\mathbf{K}^\top \mathbf{1}_{|\mathbf{Z}_{\mathcal{T}}^c|} \oslash (|\mathbf{Z}_{\mathcal{T}}^c| \cdot \mathbf{1}_n)\right)^{-1}$
 8:     Increment iteration counter $t \leftarrow t + 1$
 9: **end while**
10: Final transport matrix: $\mathbf{P}^{\lambda_1} \leftarrow \mathbf{K}$
11: Compute guidance: $\mathcal{G}_{\mathrm{W}}(\mathbf{z}_t^c) \leftarrow \langle \mathbf{P}^{\lambda_1}, \mathbf{D} \rangle = \sum_{i,j} \mathbf{P}_{ij}^{\lambda_1} \mathbf{D}_{ij}$
12: **return** $\mathcal{G}_{\mathrm{W}}(\mathbf{z}_t^c)$

---

**Algorithm 3** Computation of Batch-wise OT for Student Logit Matching

---

**Require:** Teacher output $\mathbf{t}$, Student output $\mathbf{s}$,
    Hyper-parameter $\lambda_2$, Maximum number of iterations $T$
**Ensure:** Sinkhorn loss $\mathcal{L}_{\text{SD}}$
  1: Apply softmax: $\mathbf{t} \leftarrow \text{Softmax}(\mathbf{t}), \quad \mathbf{s} \leftarrow \text{Softmax}(\mathbf{s})$
  2: Compute distance matrix $\mathbf{C}_{ij}(\mathbf{t}, \mathbf{s}) = \| \mathbf{t}(\mathbf{x}_i) - \mathbf{s}(\mathbf{x}_j) \|_p$
  3: Compute kernel matrix $\mathbf{K} \leftarrow \exp\left(-\frac{\mathbf{C}}{\lambda_2}\right)$
  4: Set iteration counter $t \leftarrow 0$
  5: **while** $t < T$ **do**
  6:     Row normalization: $\mathbf{K} \leftarrow \mathbf{K} \oslash \left(\mathbf{K}\mathbf{1}_b\mathbf{1}_b^{\text{T}}\right)$
  7:     Column normalization: $\mathbf{K} \leftarrow \mathbf{K} \oslash \left(\mathbf{1}_b\mathbf{1}_b^{\text{T}}\mathbf{K}\right)$
  8:     Increment iteration counter $t \leftarrow t + 1$
  9: **end while**
10: Sinkhorn loss $\mathcal{L}_{\text{SD}} \leftarrow \langle \mathbf{K}, \mathbf{C} \rangle = \sum_{i,j} \mathbf{K}_{ij}\mathbf{C}_{ij}$
11: **return** $\mathcal{L}_{\text{SD}}$

---

**Algorithm 4** Class-wise OT Distance in Label–Image Space for Contraction Factor $\alpha$ Calculation

---

**Require:** Real latent sets $\mathbf{Z}_{\mathcal{T}} \in \mathbb{R}^{N_1 \times d}$, distilled latent sets $\mathbf{Z}_{\mathcal{S}} \in \mathbb{R}^{N_2 \times d}$;
    One-hot labels $\mathbf{H}_{\mathcal{T}} \in \{0, 1\}^{N_1 \times C}$, soft labels $\mathbf{S}_{\mathcal{S}} \in [0, 1]^{N_2 \times C}$;
    Regularization parameter $\varepsilon > 0$, iterations $T$
**Ensure:** Average classwise OT distance $\mathcal{L}_{\text{avg}}$
  1: Compute pairwise cost matrix $\mathbf{C}_{ij} \leftarrow \|\mathbf{Z}_{\mathcal{T}}(i) - \mathbf{Z}_{\mathcal{S}}(j)\|_p$
  2: Initialize list of valid class distances W
  3: **for** each class $c = 1$ to $C$ **do**
  4:     $\tilde{\mathbf{a}} \leftarrow \mathbf{H}_{\mathcal{T}}[:, c], \quad \tilde{\mathbf{b}} \leftarrow \mathbf{S}_{\mathcal{S}}[:, c]$
  5:     **if** $\sum_i \tilde{\mathbf{a}}_i = 0$ **or** $\sum_j \tilde{\mathbf{b}}_j = 0$ **then**
  6:         **continue**
  7:     **end if**
  8:     $\mathbf{a} \leftarrow \tilde{\mathbf{a}}/\sum_i \tilde{\mathbf{a}}_i$
  9:     $\mathbf{b} \leftarrow \tilde{\mathbf{b}}/\sum_j \tilde{\mathbf{b}}_j$
10:     $\mathbf{K} \leftarrow \exp(-\mathbf{C}/\varepsilon)$
11:     Initialize $\mathbf{u} \leftarrow \mathbf{1}/N_1$
12:     Initialize $\mathbf{v} \leftarrow \mathbf{1}/N_2$
13:     **for** $t = 1$ to $T$ **do**
14:         $\mathbf{u} \leftarrow \mathbf{a}/(\mathbf{K} \cdot \mathbf{v} + \delta)$
15:         $\mathbf{v} \leftarrow \mathbf{b}/(\mathbf{K}^\top \cdot \mathbf{u} + \delta)$
16:     **end for**
17:     $\gamma \leftarrow \text{diag}(\mathbf{u}) \cdot \mathbf{K} \cdot \text{diag}(\mathbf{v})$
18:     $\mathcal{L}_c \leftarrow \sum_{i,j} \gamma_{ij}\mathbf{C}_{ij}$
19:     Append $\mathcal{L}_c$ to list $\mathcal{L}$
20: **end for**
21: W $\leftarrow \frac{1}{|\mathcal{L}|}\sum_c \mathcal{L}_c$
22: **return** W

---

# F  Implementation Details

To ensure a fair and rigorous evaluation, we adopt the training protocols and experimental configurations established by IGD [24] and EDC [23], maintaining full consistency in model architecture, optimization settings, and evaluation pipelines. Following Minimax [45] and IGD, we utilize a latent DiT model from Pytorch's official repository and an open-source VAE model from Stable Diffusion. DDIM [91] with 50 denoised steps is used as the vanilla sampling method for generation. Also, all hyperparameters related to trajectory and diversity guidance are directly inherited from IGD, while the settings for student model logit matching follow those of EDC, with the exception of parameters introduced by our optimal transport (OT) framework. Notably, most of the OT-specific hyperparameters are set to fixed values across all datasets, and we observe that they require minimal tuning to achieve strong performance. This demonstrates the robustness of our method and its low sensitivity to OT parameter variations. Comprehensive hyperparameter configurations for all benchmark datasets including ImageNet-1K, ImageNette, ImageNet-100, and CIFAR-100 are detailed in Tables 13, 14, 15, and 16, respectively.

Table 13: Hyperparameter setting on ImageNet-1K [22].

| Config | Value | Explanation |
|---|---|---|
| | | Guided Diffusion Sampling |
| $k$ | 5 | $\rho_t = k \cdot \sqrt{1-\alpha_t} \cdot \frac{\|\epsilon_\phi(\mathbf{z}_t,t,c)\|}{\|\nabla_{\mathbf{z}_t}\mathcal{G}_I(\hat{\mathbf{z}}_0|t)\|}$ |
| $\gamma_t$ | 120 | Weight for Diversity Guidance |
| $\beta_1$ | 1 | Weight for OT Sampling Guidance |
| $\lambda_1$ | 1000 | Entropy Regularization Weight |
| $T$ | 20 | Sinkhorn Iterations, Same for Logit Matching |
| | | Soft Label Relabeling |
| Epochs | 300, 500, 1000 | 300 for comparison with most baselines |
| Batch Size | 50 | Use 100 when IPC = 50 |
| $\mathbb{T}(\text{IPC} = 10)$ | ResNet-18, ShuffleNet | NA |
| $\mathbb{T}(\text{IPC} = 50)$ | ResNet-18, MobileNet, EfficientNet, ShuffleNet | NA |
| | | Student Model Logit Matching |
| Optimizer | AdamW | NA |
| Learning Rate | 0.001 | Only use 1e-4 for Swin-Transformer |
| EMA Rate | 0.99 | Control EMA-based Evaluation |
| $\kappa_1, \kappa_2$ | 1, 0.025 | Inherit from EDC |
| $\beta_2$ | 0.1 | Weight for $\mathcal{L}_{\text{SD}}$ |
| $\lambda_2$ | 0.1 | Entropy Regularization Weight |
| Scheduler | Smoothing LR Schedule | $\zeta = 2$ |
| Augmentation | RandomResizedCrop RandomHorizontalFlip | NA |

# G  Further Experimental Analyses

## G.1  More Sensitivity Analysis

In the main text, we presented a preliminary sensitivity analysis of key hyperparameters. To further evaluate their impact on performance, we conduct extensive ablation studies, with results summarized in Figures 4, 5, 6, 7 and Table 17. Overall, we observe that our method exhibits strong robustness to most hyperparameter settings: performance remains stable across a broad range of values. We select the hyperparameters by considering trade-offs among different architectures. Specifically, increasing the value of $\beta_1$ slightly improves the performance of ConvNet, but degrades that of ResNet-18. We therefore set $\beta_1 = 1$ to balance this trade-off. Similarly, increasing $\beta_2$ enhances performance on ResNet-AP, but negatively affects both ResNet-18 and ConvNet. Thus, we choose $\beta_2 = 0.1$ to

Table 14: Hyperparameter setting on ImageNette [49].

| Config | Value | Explanation |
|---|---|---|
| | Guided Diffusion Sampling | |
| $k$ | 5 | $\rho_t = k \cdot \sqrt{1-\alpha_t} \cdot \dfrac{\|\epsilon_\phi(\mathbf{z}_t,t,c)\|}{\|\nabla_{\mathbf{z}_t}\mathcal{G}_I(\hat{\mathbf{z}}_0\mid t)\|}$ |
| $\gamma_t$ | 50 when IPC=10
120 when IPC=50 or 100 | Weight for Diversity Guidance |
| $\beta_1$ | 1 | Weight for OT Sampling Guidance |
| $\lambda_1$ | 1000 when IPC=10
3000 when IPC=50 or 100 | Entropy Regularization Weight |
| $T$ | 20 | Sinkhorn Iterations, Same for Logit Matching |
| | Soft Label Relabeling | |
| Epochs | 1000 | Same for reproducing IGD |
| Batch Size | 50 when IPC=10
100 when IPC=50 or 100 | NA |
| $\mathbb{T}(\text{IPC}=10)$ | ResNet-18, MobileNet | NA |
| $\mathbb{T}(\text{IPC}=100)$ | ResNet-18, MobileNet,
EfficientNet, ShuffleNet | Same for IPC=50 |
| | Student Model Logit Matching | |
| Optimizer | AdamW | NA |
| Learning Rate | 0.001 | NA |
| EMA Rate | 0.99 | Control EMA-based Evaluation |
| $\kappa_1, \kappa_2$ | 1, 0.025 | Inherit from EDC |
| $\beta_2$ | 0.1 | Weight for $\mathcal{L}_{\text{SD}}$ |
| $\lambda_2$ | 0.1 | Entropy Regularization Weight |
| Scheduler | Smoothing LR Schedule | $\zeta = 2$ |
| Augmentation | RandAugment
RandomResizedCrop
RandomHorizontalFlip | NA |

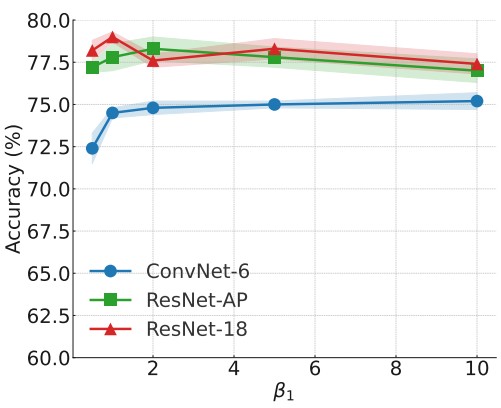

Figure 4: Effect of $\beta_1$ (OT sampling weight) on ImageNette [49] (IPC=10).

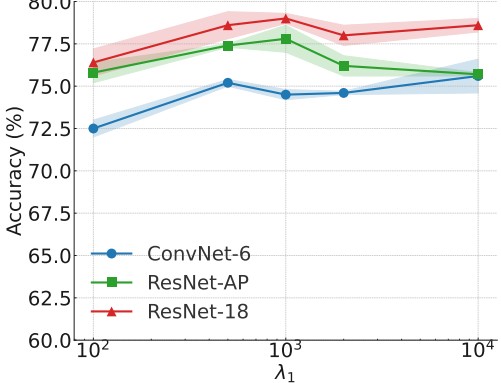

Figure 5: Effect of $\lambda_1$ (entropy regularization weight for sampling) on ImageNette [49] (IPC=10).

achieve the best average performance. In contrast, reducing either $\beta_1$ or $\beta_2$ consistently harms all model variants, which highlights the importance and effectiveness of our OT-based regularization terms. Since the latent and feature spaces differ in scale, we apply separate scaling factors $\lambda_1$ and $\lambda_2$ to normalize their contributions. As shown in the figures, setting $\lambda_1 = 1000$ and $\lambda_2 = 0.1$ yields a favorable trade-off across architectures. Taken together, our approach demonstrates two complementary aspects of robustness: (1) most OT-related hyperparameters exhibit consistent

Table 15: Hyperparameter setting on ImageNet-100 [13].

| Config | Value | Explanation |
|--------|-------|-------------|
| | Guided Diffusion Sampling | |
| $k$ | 5 | $\rho_t = k \cdot \sqrt{1-\alpha_t} \cdot \frac{\|\epsilon_\phi(\mathbf{z}_t,t,c)\|}{\|\nabla_{\mathbf{z}_t}\mathcal{G}_I(\hat{\mathbf{z}}_0|t)\|}$ |
| $\gamma_t$ | 120 | Weight for Diversity Guidance |
| $\beta_1$ | 1 | Weight for OT Sampling Guidance |
| $\lambda_1$ | 1000 | Entropy Regularization Weight |
| $T$ | 20 | Sinkhorn Iterations, Same for Logit Matching |
| | Soft Label Relabeling | |
| Epochs | 300 | NA |
| Batch Size | 100 | NA |
| $\mathbb{T}(\text{IPC}=10)$ | ResNet-18, ShuffleNet | NA |
| $\mathbb{T}(\text{IPC}=100)$ | ResNet-18, MobileNet, EfficientNet, ShuffleNet | Same for IPC=50 |
| | Student Model Logit Matching | |
| Optimizer | AdamW | NA |
| Learning Rate | 0.001 | NA |
| EMA Rate | 0.99 | Control EMA-based Evaluation |
| $\kappa_1, \kappa_2$ | 1, 0.025 | Inherit from EDC |
| $\beta_2$ | 0.1 | Weight for $\mathcal{L}_{\text{SD}}$ |
| $\lambda_2$ | 0.1 | Entropy Regularization Weight |
| Scheduler | Smoothing LR Schedule | $\zeta = 2$ |
| Augmentation | RandomResizedCrop RandomHorizontalFlip | NA |

Table 16: Hyperparameter setting on CIFAR-100 [21].

| Config | Value | Explanation |
|--------|-------|-------------|
| | Guided Diffusion Sampling | |
| $k$ | 5 | $\rho_t = k \cdot \sqrt{1-\alpha_t} \cdot \frac{\|\epsilon_\phi(\mathbf{z}_t,t,c)\|}{\|\nabla_{\mathbf{z}_t}\mathcal{G}_I(\hat{\mathbf{z}}_0|t)\|}$ |
| $\gamma_t$ | 120 | Weight for Diversity Guidance |
| $\beta_1$ | 1 | Weight for OT Sampling Guidance |
| $\lambda_1$ | 1000 | Entropy Regularization Weight |
| $T$ | 20 | Sinkhorn Iterations, Same for Logit Matching |
| | Soft Label Relabeling | |
| Epochs | 1000 | NA |
| Batch Size | 50 | NA |
| $\mathbb{T}(\text{IPC}=10)$ | ResNet-18, ShuffleNet | NA |
| $\mathbb{T}(\text{IPC}=100)$ | ResNet18, ConvNet, MobileNet, WRN, ShuffleNet | Same for IPC=50 |
| | Student Model Logit Matching | |
| Optimizer | AdamW | NA |
| Learning Rate | 0.001 | NA |
| EMA Rate | 0.99 | Control EMA-based Evaluation |
| $\kappa_1, \kappa_2$ | 1, 0.025 | Inherit from EDC |
| $\beta_2$ | 0.1 | Weight for $\mathcal{L}_{\text{SD}}$ |
| $\lambda_2$ | 0.1 | Entropy Regularization Weight |
| Scheduler | Smoothing LR Schedule | $\zeta = 2$ |
| Augmentation | RandAugment RandomResizedCrop RandomHorizontalFlip | NA |

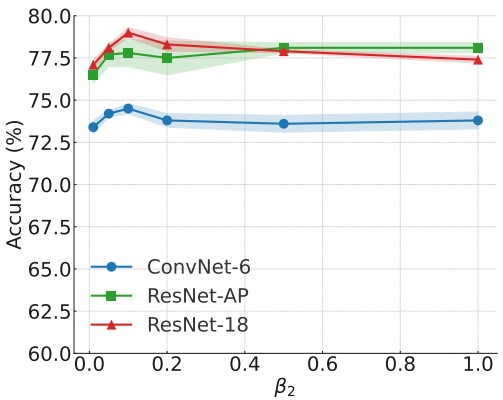
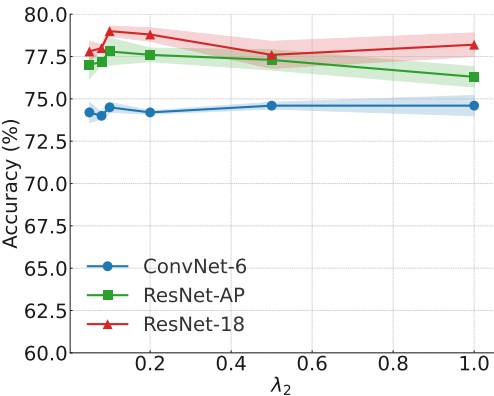

Figure 6: Effect of $\beta_2$ (OT matching weight) on ImageNette [49] (IPC=10).

Figure 7: Effect of $\lambda_2$ (entropy regularization weight for logit matching) on ImageNette [49] (IPC=10).

Table 17: Effect of $T$ on ImageNette [49] (IPC=10).

| $T$ | 5 | 10 | 20 | 50 | 100 |
|-----------|------|------|------|------|------|
| ResNet-18 | 77.9 | 78.6 | 79.0 | 78.8 | 78.9 |
| ConvNet-6 | 73.8 | 74.3 | 74.5 | 74.7 | 74.6 |

behavior across different scenarios, requiring little to no manual adjustment, and in practice we only select $\lambda_1$ from the set $\{1000, 3000\}$ while keeping all other OT-related hyperparameters fixed (see Section F for default values); and (2) performance remains stable even when these parameters vary within reasonable ranges, eliminating the need of careful tuning.

### G.2 Further Analysis of the Contraction Factor $\alpha$

We provide a formal characterization and empirical analysis of the contraction factor $\alpha$, which quantifies the degree to which soft labels reduce the discrepancy between the label and image distributions compared to hard labels. This factor plays a critical role in interpreting the effectiveness of soft supervision in dataset distillation. For efficient computation, we approximate the contraction factors using features in the latent space, enabling dimensionality reduction while preserving critical information.

**Definition and Computation**    To compute $\alpha$, we compare the class-conditional optimal transport (OT) distances from the real dataset to two variants of the distilled dataset: one annotated with soft labels $\nu_{\text{distill}}^{(\text{soft})}$ and one with hard labels $\nu_{\text{distill}}^{(\text{hard})}$. The contraction factor is then defined as the relative improvement in transport distance under soft supervision: $\alpha = \text{W}(\mu_{\text{true}}, \nu_{\text{distill}}^{(\text{soft})}) / \text{W}(\mu_{\text{true}}, \nu_{\text{distill}}^{(\text{hard})})$.

Let $\mathbf{Z}_{\mathcal{T}} \in \mathbb{R}^{N_1 \times d}$ and $\mathbf{Z}_{\mathcal{S}} \in \mathbb{R}^{N_2 \times d}$ denote latent embeddings extracted from real and distilled images, respectively. We construct the pairwise cost matrix using an $\ell_p$ norm:

$$\mathbf{C}_{ij} = \|\mathbf{Z}_{\mathcal{T}}(i) - \mathbf{Z}_{\mathcal{S}}(j)\|_p, \quad \mathbf{C} \in \mathbb{R}^{N_1 \times N_2}. \tag{19}$$

For each class $c \in \{1, \ldots, C\}$, we extract the marginal label distributions over samples: $\tilde{\mathbf{a}}^{(c)} = \mathbf{H}_{\mathcal{T}}[:, c]$ for real hard labels, and $\tilde{\mathbf{b}}^{(c)} = \mathbf{S}_{\mathcal{S}}[:, c]$ for distilled soft labels, where $\mathbf{H}_{\mathcal{T}} \in \{0, 1\}^{N_1 \times C}$ and $\mathbf{S}_{\mathcal{S}} \in [0, 1]^{N_2 \times C}$. These are normalized into valid probability vectors:

$$\mathbf{a}^{(c)} = \frac{\tilde{\mathbf{a}}^{(c)}}{\sum_i \tilde{a}_i^{(c)}}, \quad \mathbf{b}^{(c)} = \frac{\tilde{\mathbf{b}}^{(c)}}{\sum_j \tilde{b}_j^{(c)}}. \tag{20}$$

We then perform entropic regularized OT using the Sinkhorn algorithm. The Gibbs kernel is defined as:

$$\mathbf{K} = \exp\left(-\frac{\mathbf{C}}{\varepsilon}\right), \tag{21}$$

Table 18: Effect of $\alpha$ on ImageNet-1K [22] (IPC=10). **Config A:** ResNet-18. **Config B:** ResNet-18, MobileNet, EfficientNet, ShuffleNet. **Config C:** ResNet-18, MobileNet, AlexNet, ShuffleNet. **Config D:** ResNet-18, ShuffleNet.

| Teachers | Config A | Config B | Config C | Config D |
|---|---|---|---|---|
| $\alpha$ | 1.00 | 0.99 | 0.95 | 0.93 |
| Acc (ResNet-18) | 50.3 | 52.3 | 52.7 | 52.9 |
| Acc (Swin) | 47.2 | 47.8 | 49.2 | 50.2 |

Table 19: Effect of $\alpha$ on ImageNet-1K [22] (IPC=50). **Config A:** ResNet-18. **Config B:** ResNet-18, MobileNet, EfficientNet, ShuffleNet. **Config C:** ResNet-18, MobileNet, AlexNet, ShuffleNet. **Config D:** ResNet-18, ShuffleNet.

| Teachers | Config A | Config B | Config C | Config D |
|---|---|---|---|---|
| $\alpha$ | 0.97 | 0.16 | 0.97 | 1.00 |
| Acc (ResNet-18) | 62.3 | 61.9 | 60.8 | 60.5 |
| Acc (Swin) | 65.5 | 68.2 | 65.5 | 65.3 |

where $\varepsilon$ controls regularization strength. The scaling vectors $\mathbf{u}$ and $\mathbf{v}$ are initialized uniformly as:

$$\mathbf{u}^0 \leftarrow \mathbf{1}/N_1, \quad \mathbf{v}^0 \leftarrow \mathbf{1}/N_2, \tag{22}$$

and iteratively updated as:

$$\mathbf{u}^{t+1} = \frac{\mathbf{a}^{(c)}}{\mathbf{K}\mathbf{v}^t + \delta}, \tag{23}$$

$$\mathbf{v}^{t+1} = \frac{\mathbf{b}^{(c)}}{\mathbf{K}^\top \mathbf{u}^{t+1} + \delta}, \tag{24}$$

where $\delta$ ensures numerical stability. After $T$ iterations, the transport plan is:

$$\gamma^{(c)} = \text{diag}(\mathbf{u}) \cdot \mathbf{K} \cdot \text{diag}(\mathbf{v}), \tag{25}$$

and the classwise OT cost becomes:

$$\mathcal{L}_c = \langle \gamma^{(c)}, \mathbf{C} \rangle = \sum_{i,j} \gamma_{ij}^{(c)} C_{ij}. \tag{26}$$

Averaging over the valid class set $\mathcal{C}$ (i.e., classes with non-zero support in both distributions) yields:

$$W(\mu_{\text{true}}, \nu_{\text{distill}}^{(\text{soft})}) = \frac{1}{|\mathcal{C}|} \sum_{c \in \mathcal{C}} \mathcal{L}_c. \tag{27}$$

To compute the counterpart $W(\mu_{\text{true}}, \nu_{\text{distill}}^{(\text{hard})})$, we replace $\mathbf{S}_{\mathcal{S}}$ with its hard label projection $\mathbf{S}_{\mathcal{S}}^h \in \{0, 1\}^{N_2 \times C}$ and repeat the same computation.

**Empirical Insights** We conduct several additional experiments, with results shown in Tables 18 and 19. Our empirical analysis provides several important observations regarding the role of the contraction factor $\alpha$ in guiding effective distillation. We first find that $\alpha$ is highly sensitive to the diversity and calibration quality of the teacher ensemble, and this sensitivity is modulated by the IPC (images per class) setting. In low-IPC regimes (e.g., IPC=10), using overly complex or inconsistent teacher predictions increases the optimal transport distance $W(\mu_{\text{true}}, \nu_{\text{distill}}^{(\text{soft})})$, leading to smaller $\alpha$ values and ultimately harming the generalization ability of the distilled dataset. Conversely, when IPC is sufficiently high (e.g., IPC=50), stronger and more expressive teacher distributions better capture the semantic structure of the real data, resulting in larger $\alpha$ values and improved alignment between real and synthetic distributions.

Second, we observe that deliberately reducing $\alpha$, thereby explicitly minimizing the overall optimal transport distance, leads to significant improvements in downstream model performance. This effect is particularly evident under both settings, where configurations with smaller $\alpha$ (e.g., Config D)

Table 20: Comparison of our optimal transport-based distance with other measures for guided diffusion sampling on ImageNet-1K [22].

| IPC | MMD | MMD (RKHS) | Ours |
|-----|-----|-----------|------|
| 10 | 49.4 | 50.3 | 52.9 |
| 50 | 60.4 | 60.6 | 61.9 |

Table 21: Comparison between the sample-wise and batch-wise optimal transport distance for student model logit matching on ImageNet-1K [22]. OOM: CUDA out of memory.

| Level | ResNet-18 | | MobileNet-V2 | |
|-------|-----------|-----------|--------------|-----------|
| | IPC=10 | IPC=50 | IPC=10 | IPC=50 |
| Sample-wise | 50.6 | 60.8 | OOM | OOM |
| Batch-wise | 52.9 | 61.9 | 51.0 | 61.0 |

achieve better Top-1 accuracy across both the ResNet-18 and Swin Transformer. These results empirically confirm that shrinking the distributional gap through controlling $\alpha$ facilitates more efficient and effective knowledge transfer. Third, in high-IPC regimes, when a single teacher is used (e.g., Config A), student models that share the same architecture as the teacher can fully exploit the teacher's architectural biases, achieving strong performance. However, such tight alignment may limit generalization to unseen architectures. By appropriately contracting $\alpha$, we encourage the distilled dataset to encode more transferable, architecture-independent features, thereby improving the student's adaptability to diverse downstream architectures. Overall, these findings validate $\alpha$ as a principled and tunable indicator of distillation quality, and highlight the importance of strategic contraction strategies tailored to both teacher complexity and downstream generalization targets.

### G.3 Comparison with Other Distance Measure

**For guided diffusion sampling**    Unlike conventional metrics such as cosine similarity, KL divergence, or mean squared error, which rely on explicit instance-level alignment, optimal transport (OT) enables distribution-level alignment without enforcing one-to-one correspondences. This property makes OT particularly well-suited for dataset distillation scenarios, where the number of synthetic samples is significantly smaller than that of the original dataset, and direct pairing is often infeasible or suboptimal. While Maximum Mean Discrepancy (MMD)-based measures have also been adopted for distribution alignment without requiring exact correspondences, they primarily focus on matching global distributional statistics and fail to capture fine-grained pairwise relations between individual instances in the real and synthetic distributions. In contrast, our OT-based formulation explicitly models such pairwise interactions and thus facilitates more accurate and semantically consistent guidance during the diffusion sampling process. As shown in Table 20, our method consistently outperforms MMD and MMD with reproducing kernel Hilbert spaces (RKHS) baselines on ImageNet-1K under both low-IPC and high-IPC settings. These results underscore the importance of modeling instance-level correspondences for effective guidance and highlight the superiority of OT in capturing the geometry of complex data distributions.

**For student model logit matching**    Table 21 illustrates the consistent superiority of batch-wise OT distance over sample-wise OT distance. This result highlights that batch-wise Sinkhorn distance is more effective in transferring the distributional geometry captured by the distilled set from label-image space to the newly trained student models. The sample-wise logit matching approach treats each instance independently, failing to account for the global structure and correlations within a batch.

Table 22: Performance comparison of logit matching methods on ImageNette (IPC=10).

| Network | ConvNet-6 | ResNetAP-10 | ResNet-18 |
|---------|-----------|-------------|-----------|
| MMD | 72.6 | 75.3 | 76.4 |
| KL | 73.0 | 75.4 | 77.6 |
| OTM | **74.5** | **77.8** | **79.0** |

Table 23: Distribution coverage comparison among different methods.

| Threshold | DiT [44] | DiT-IGD [24] | Ours |
|---|---|---|---|
| 10 | 40.2 | 40.8 | 41.6 |
| 12 | 54.6 | 56.3 | 57.5 |

In contrast, our batch-wise formulation preserves inter-sample relationships, enabling more faithful distributional alignment and resulting in more robust knowledge transfer. Moreover, when dealing with datasets containing a large number of classes (e.g., 1,000 classes in ImageNet-1K), the batch-wise approach substantially reduces memory consumption and avoids the CUDA out-of-memory issues frequently encountered by sample-wise matching, further enhancing its scalability.Also, although KL and MMD serve as simpler divergences, they are inherently limited. KL divergence is applied per sample and ignores inter-sample relationships, while MMD matches only global statistics. In contrast, our OTM applies batch-wise OT alignment between student logits and soft labels, capturing the joint distributional structure of samples. This enables OT to faithfully preserve inter-sample geometry and match structural uncertainty in the soft labels, which KL and MMD overlook. As shown in Table 22, this leads to a clear performance gain:

### G.4 More Discussions on Runtime

While EDC [23] reduces the runtime during the recovery phase compared to previous work, it does not fully optimize for multi-GPU parallelism across its various processes. Specifically, the pre-sampling and post-sampling phases during initialization do not benefit from multi-GPU parallelism, as parallelizing these steps does not result in substantial time reduction. Moreover, the recovery phase is inherently constrained by data loading and model-inversion methods, and beyond four GPUs, further increases in parallelism yield minimal improvements in runtime. In contrast, our approach is designed to optimize each class separately, with the sampling process dependent only on images sampled from the same class and the corresponding real images. As a result, our method scales more efficiently with the number of GPUs, with runtime decreasing nearly inversely proportional to the number of GPUs. Furthermore, when the need for high IPC arises, our method can be adapted to split high-IPC tasks into several lower-IPC ones for parallel processing, maintaining strong parallel efficiency and further enhancing its applicability in real-world scenarios.

### G.5 Data Coverage Analysis

To assess the representational fidelity of the distilled dataset, we adopt a coverage-based evaluation metric. Specifically, for each data point in the original dataset, we determine whether it has at least one nearest neighbor in the distance dataset within a predefined distance threshold. This metric reflects how well the surrogate data captures the underlying structure of the original distribution. As shown in Table 23, our method consistently achieves higher coverage compared to baseline methods across multiple thresholds. The improvements are observed over both the original DiT [44] model and DiT-IGD [24], indicating that our approach provides better distributional alignment. Notably, the performance gap widens as the threshold increases, further validating the robustness of our distilled data in covering diverse modes of the original dataset.

### G.6 Comparison with DWA

DWA [92] enhances diversity by adjusting the statistics of the squeezed network based on each generated sample. However, it still relies solely on global statistics, specifically the mean and variance associated with batch normalization (BN), and thus fails to capture the rich instance-level information and geometric distributional structures inherent in the real dataset. Visualizations from the DWA paper further illustrate that, while the directed weight adjustment improves the diversity of the distilled dataset, the distribution remains concentrated, failing to adequately cover the majority of the real data distribution. In Table 24, we compare our method with DWA across multiple student models, and the results clearly demonstrate a significant performance advantage of our approach. This further emphasizes the importance of leveraging fine-grained instance-level information for achieving improved model performance and more faithful distributional alignment.

Table 24: Comparison with DWA [92] on ImageNet-1K [22].

| Method | ResNet-18 | | MobileNet-V2 | | EfficientNet-B0 | |
| | IPC10 | IPC50 | IPC10 | IPC50 | IPC10 | IPC50 |
|---|---|---|---|---|---|---|
| DWA [92] | 37.9±0.2 | 55.2±0.2 | 29.1±0.3 | 51.6±0.5 | 37.4±0.5 | 56.3±0.4 |
| Ours | **52.9±0.1** | **61.9±0.5** | **51.0±0.6** | **61.0±0.4** | **56.7±0.2** | **64.4±0.1** |

Table 25: Comparison between WMDD [42] and our method on ImageNette [49] and ImageNet-1K [22] under different IPC settings.

| Dataset | ImageNette | | ImageNet-1K | |
| IPC | 10 | 50 | 10 | 50 |
|---|---|---|---|---|
| WMDD | 64.8±0.4 | 83.5±0.3 | 38.2±0.2 | 57.6±0.5 |
| Ours | **79.0±0.3** | **89.3±0.3** | **52.9±0.1** | **61.9±0.5** |

## G.7 Comparison with WMDD

Although both our method and WMDD [42] utilize optimal transport (OT), they differ significantly in both methodology and motivation, leading to distinct formulations and implementations.

WMDD [42] is a distribution-matching-based distillation method that applies OT in a single, offline step to compute a Wasserstein barycenter over the real data's feature distribution. This barycenter is then used as a fixed target throughout training, where synthetic images are optimized to match it using a standard L2 loss in the feature space. In contrast, we introduce a fundamentally different generative paradigm where OT is not a static, one-off computation, but a dynamic guidance mechanism integrated throughout the entire data synthesis and training pipeline. Specifically, OT guides the sampling of synthetic images by aligning latent representations, regulates the soft label relabeling process by matching label complexity to the image distribution, and structures the training loss of the student model by aligning its logits to the relabeled targets.

The motivation behind WMDD is to replace the use of simple data summaries, such as the feature means often targeted by MMD-based methods, with a more geometrically meaningful summary, namely the Wasserstein barycenter, derived from the Wasserstein metric. In contrast, our method is driven by the need to address inherent limitations in generative distillation pipelines, which often fail to preserve the fine-grained geometry of the real data distribution—particularly intra-class variations and local modes. These aspects are explicitly addressed in our framework through a multi-stage, OT-guided design.

We compare the top-1 accuracy of our method with WMDD under different images-per-class (IPC) settings on both the ImageNette and ImageNet-1K datasets in Table 25.

## G.8 Robustness Evaluation

To assess the robustness of student models trained with distilled datasets, we follow the evaluation protocol established in DD-RobustBench [93], utilizing adversarial attacks implemented in the TorchAttacks library [94]. As shown in Tables 26 and 27, we evaluate models trained on ImageNette [49] under IPC=10 and IPC=50 settings, measuring both standard test accuracy and adversarial robustness against a variety of attack methods.

Our method consistently achieves higher clean accuracy and substantially improves robustness compared to MTT [14] across different perturbation budgets ($|\varepsilon| = 4/255$ and $|\varepsilon| = 8/255$). These improvements can be attributed to the distributional properties enforced by our optimal transport (OT)-based distillation framework. By minimizing the OT distance between the synthetic and real data distributions, our method preserves not only class-level statistics but also fine-grained, instance-level geometric structures. This leads to the learning of semantically faithful and smoother decision boundaries, which are inherently more resilient to adversarial perturbations. Moreover, our OT-guided diffusion sampling produces visually more coherent and perceptually realistic images compared to other types of approaches. The generated synthetic samples better preserve the semantic integrity and natural variability of the original data, providing stronger perceptual signals during model training.

As a result, the student model benefits from a more robust feature space that aligns well with human perception, further enhancing adversarial robustness beyond purely decision-boundary-level effects.

In contrast, methods that primarily match global statistics or rely on heuristic trajectory guidance, such as MTT, often produce synthetic datasets lacking such structural fidelity, resulting in brittle decision boundaries that are more vulnerable to attacks.

From a theoretical perspective, prior works [95, 96] have established a strong connection between adversarial robustness and the sharpness of decision boundaries: sharper, more irregular boundaries tend to amplify adversarial vulnerability, whereas flatter, smoother boundaries promote robustness. By aligning not only global distributions but also the local transportation cost between real and synthetic samples, OT encourages the distilled student model to form flatter and more coherent decision surfaces aligned with the real data geometry.

Moreover, from a loss landscape perspective, minimizing the OT distance guides optimization towards flatter minima, where small input perturbations induce minimal output changes. This connection is well supported by prior studies [97, 98], which show that flatter loss surfaces correlate strongly with improved adversarial robustness. Together, these empirical and theoretical insights demonstrate that preserving distributional geometry via optimal transport provides a principled and effective pathway for enhancing the adversarial robustness of models trained on distilled datasets.

Table 26: Performance comparison on DD-RobustBench [93] evaluated on ImageNette [49], under a perturbation budget of $|\varepsilon| = 4/255$. Results for MTT [14] are directly copied from the DD-RobustBench benchmark.

| Attack Methods | IPC=10 | | IPC=50 | |
|---|---|---|---|---|
| | MTT [14] | Ours | MTT [14] | Ours |
| Clean Accuracy | 66.4 | **69.1** | 67.7 | **84.6** |
| FGSM | 10.8 | **20.8** | 8.4 | **24.0** |
| PGD | 4.6 | **9.2** | 2.6 | **9.8** |
| CW | 4.6 | **12.0** | 1.4 | **14.8** |
| VMI | 5.4 | **9.0** | 2.0 | **11.2** |
| Jitter | 12.2 | **20.4** | 13.0 | **23.8** |

Table 27: Performance comparison on DD-RobustBench [93] evaluated on ImageNette [49], under a perturbation budget of $|\varepsilon| = 8/255$. Results for MTT [14] are directly copied from the DD-RobustBench benchmark.

| Attack Methods | IPC=10 | | IPC=50 | |
|---|---|---|---|---|
| | MTT [14] | Ours | MTT [14] | Ours |
| Clean Accuracy | 66.4 | **69.1** | 67.6 | **84.6** |
| FGSM | 0.8 | **11.0** | 1.8 | **14.8** |
| PGD | 0.2 | **2.8** | 1.2 | **15.0** |
| CW | 0.2 | **9.6** | 0.2 | **6.8** |
| VMI | 0.2 | **0.8** | 0.2 | **2.0** |
| Jitter | 11.4 | **12.4** | 9.8 | **14.6** |

### G.9    Evaluation on Low-IPC Settings

We have conducted additional experiments on ImageNet-1K [22] for the challenging settings of IPC=1, IPC=2, and IPC=5. The results of these new experiments are presented in the Table 28.

Importantly, in the IPC=1 setting, since only one synthetic image is generated per class, the OTG process cannot leverage previously distilled samples for alignment. Instead, for each class, we compute the OT distance between its single synthetic candidate and the corresponding real images in the latent space to guide generation.

Table 28: Performance comparison of different methods on ImageNet-1K under small IPCs (1, 2, 5). Best results are in bold.

| IPC | Method | | | | | | | |
|-----|------|-------|-------|-------|------|------|---------|------|
|     | DM   | FrePo | TESLA | SRe2L | RDED | EDC  | DiT-IGD | Ours |
| 1   | 1.5  | 7.5   | 7.7   | 0.4   | 6.6  | 12.8 | 10.7    | **15.9** |
| 2   | 1.7  | 9.7   | 10.5  | –     | 16.5 | 22.8 | 20.6    | **25.9** |
| 5   | –    | –     | –     | –     | 23.8 | 39.5 | 38.6    | **45.7** |

## H  More Visualization Results

### H.1  T-SNE Results

To assess the effectiveness of our OT-guided diffusion sampling, we present the t-SNE [99] results in Figure 8. The diversity in IGD is driven solely by cosine-similarity based diversity guidance, without leveraging the distributional structure of the real dataset. This limitation leads to insufficient coverage of critical regions in the true data distribution, such as the central region of the green (Cassette player), the lower part of the blue (Tench), and the middle-upper section of the purple (Church) areas. Consequently, several important subclasses are absent from the distilled dataset, resulting in the new model failing to learn relevant intra-class variations and important subclass-specific information. In contrast, our approach iteratively computes the optimal transport distance between the real dataset and the distilled set, explicitly incorporating both intra-class structures and finer substructures of the real data. This enables our distilled dataset to capture a broader range of essential submodalities and regions, facilitating a more comprehensive transfer of information to the new model, and minimizing information loss. By employing the optimal transport distance as an additional supervision signal during the new model's training, we ensure the effective transfer of this enriched information, leading to significant improvements in model performance.

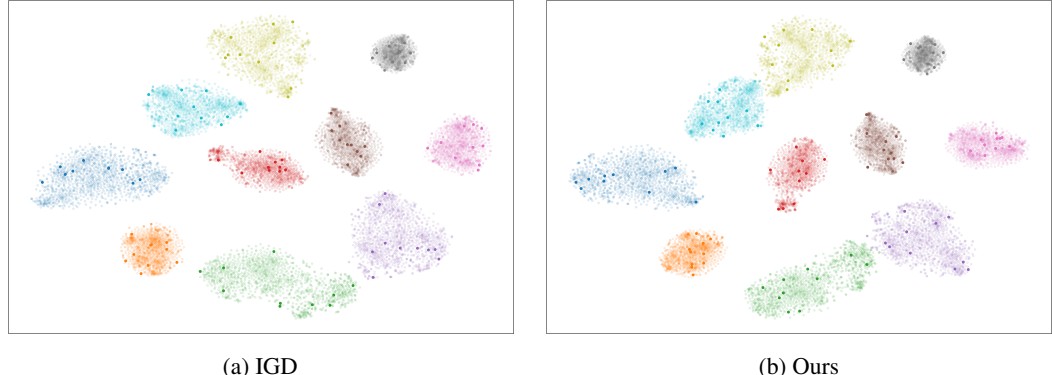

(a) IGD                                    (b) Ours

Figure 8: Visualization study for sample distributions of distilled datasets (IPC=10) generated by IGD [24] and Ours versus the original ImageNette [49] dataset. The dark points represent the distilled set, while the light points represent the real (original) set. The diversity in IGD [24] is driven solely by random diversity guidance, lacking awareness of the real data distribution. As a result, it fails to cover critical regions such as such as the central region of the green class (Cassette player), the lower part of the blue class (Tench), and the middle-upper section of the purple class (Church). In contrast, our method incorporates both intra-class structures and fine-grained substructures of the real data, which allows it to effectively cover most subclass regions.

### H.2  Distilled Images

Figures 9 and 10 provide additional visual comparisons between IGD [24] and our method, as well as standalone visualizations of our distilled dataset. Our method effectively captures the structural information of the real data distribution, resulting in high-fidelity samples with semantic diversity that faithfully reflects the underlying real-world distribution.

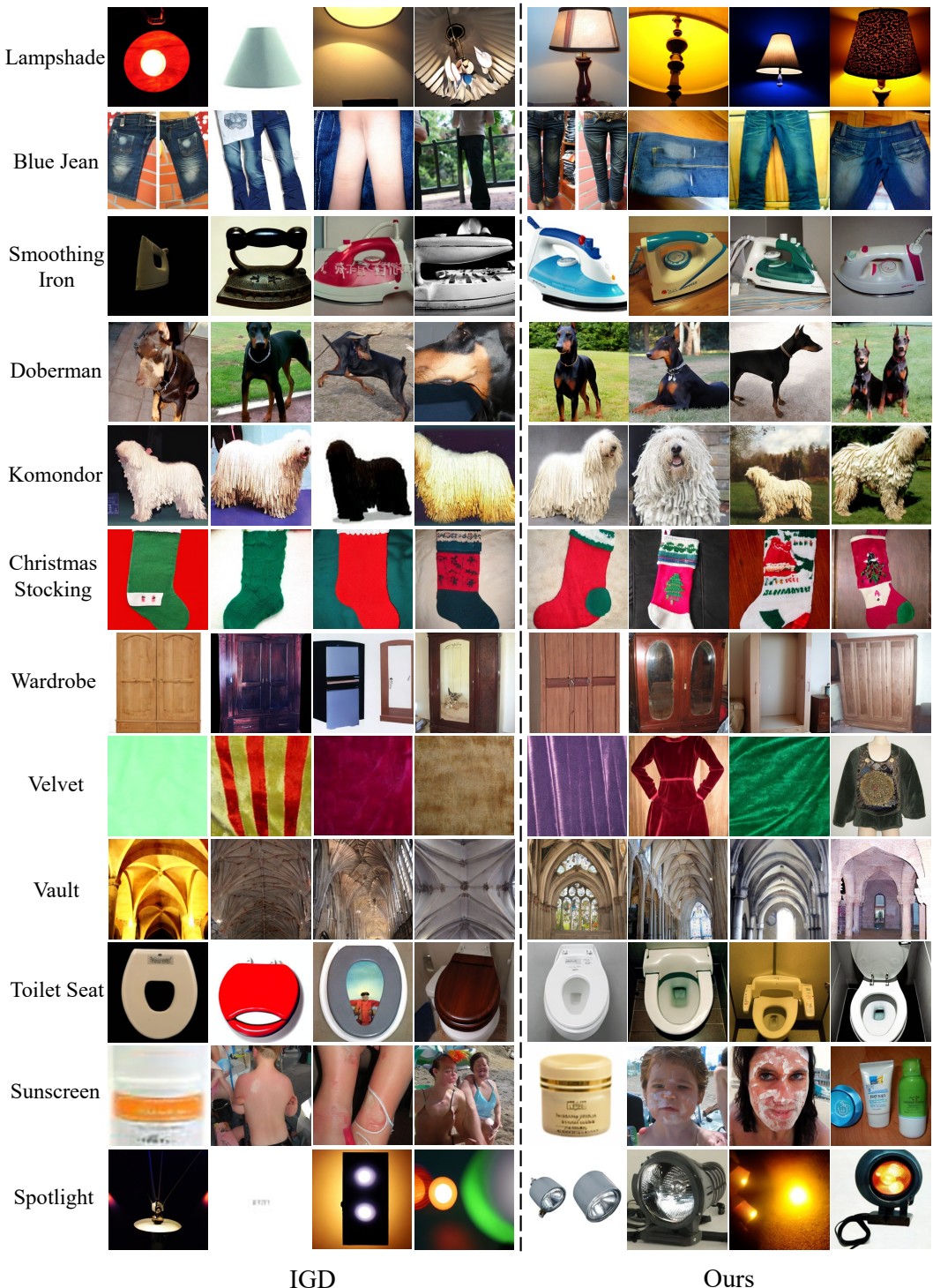

Figure 9: Additional visualization comparing the distilled datasets generated by IGD [24] and our approach on ImageNet-1K [22] (IPC=10).

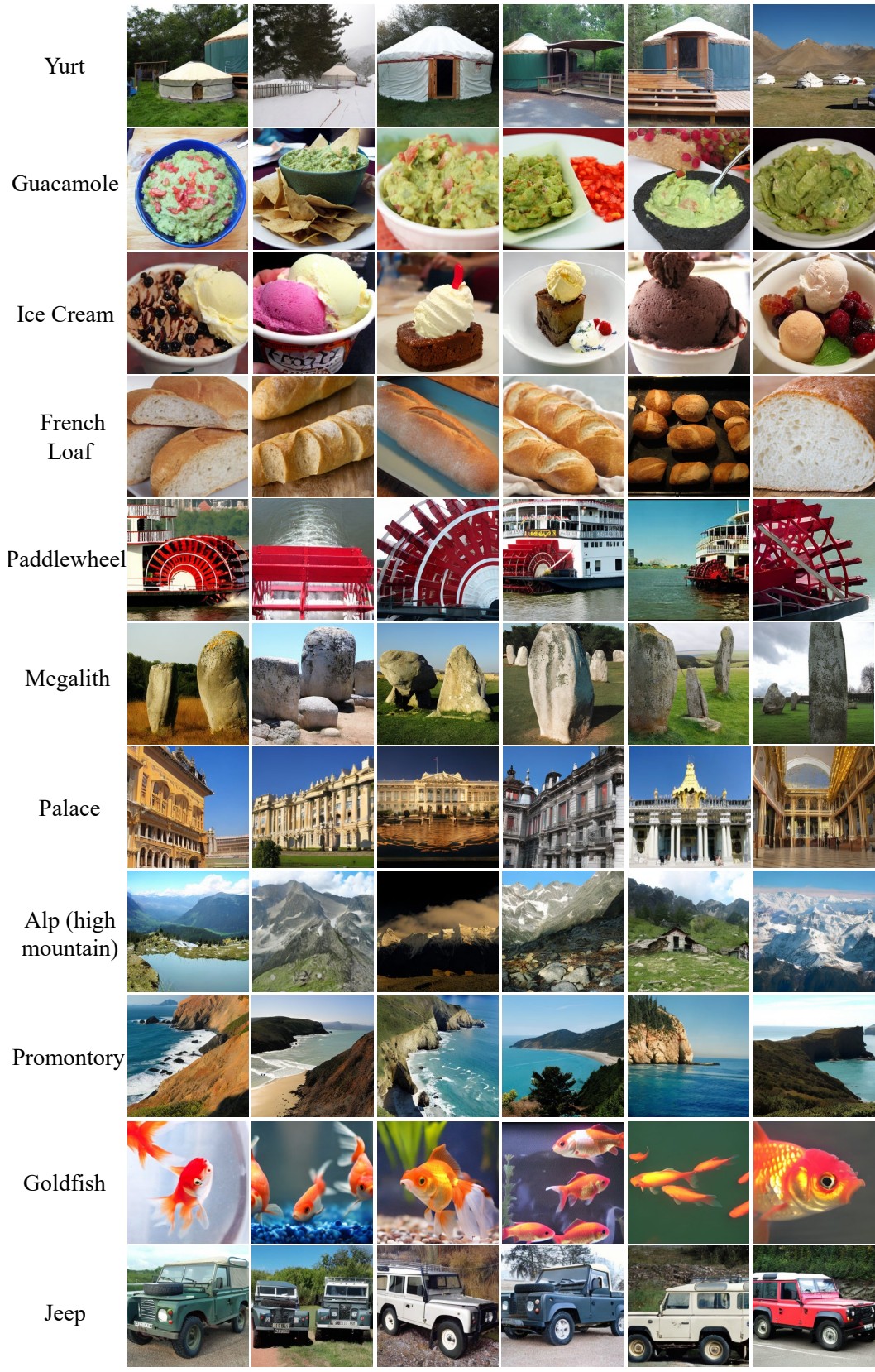

Figure 10: Additional visualizations for our distilled set on ImageNet-1K [22] (IPC=10).

# I  Limitations

Our current framework inherits the trajectory influence guidance mechanism from IGD [24], which, while effective for improving the general and global alignment of sampled data, introduces substantial computational overhead. Specifically, the additional sampling steps required to maintain trajectory consistency significantly slow down the generation process compared to the vanilla DiT [44], which operates without such constraints. In future work, we aim to reformulate the guidance process to retain benefits while reducing the reliance on explicit trajectory tracking, thereby enabling faster and more scalable sampling.

# J  Broader Impact

Our work aims to reduce dataset size while maintaining performance, enabling model training with significantly lower computational and storage costs. This can lower the entry barrier for institutions with limited resources and promote environmentally sustainable AI development [100]. Moreover, our distilled datasets have the potential to facilitate efficient learning in federated and continual learning scenarios, thereby enhancing data privacy and supporting model adaptation across distributed systems. However, as with most data-driven approaches, there exists a risk that the distilled data may retain or amplify biases present in the original datasets. This could lead to unintended consequences, particularly in sensitive applications. Additionally, by accelerating the deployment of compact models, our method may inadvertently contribute to insufficiently audited systems being widely adopted. We emphasize the importance of responsible deployment, including bias auditing, fairness-aware design, and transparency, and encourage future work to explore these aspects more thoroughly.

