# OpenReview forum: "Optimizing Distributional Geometry Alignment with Optimal Transport for Generative Dataset Distillation"
_NeurIPS.cc/2025/Conference — NeurIPS 2025 poster_

### Official Review · Reviewer_Xuuh · 2025-06-27

**Clarity:** 3
**Significance:** 3
**Originality:** 3
**Rating:** 5
**Confidence:** 4

**Summary:**

This paper proposes to incorporate optimal transport as the distance measurement between the synthetic dataset and the original dataset for dataset distillation. Specifically, the authors decompose the OT distance between these two sets into three terms and address them at different distillation stages. First, they adopt an OT-guided diffusion sampling to better maintain the structure of the original data distribution. Second, soft labels are generated based on the complexity of distilled image distributions. Last, an OT-based logit matching is employed for training student models. A detailed ablation study is conducted to illustrate the effectiveness of these three terms. Overall, the proposed method achieves state-of-the-art performance across multiple benchmarks with a marginal computational burden attached.

**Questions:**

1. What is the difference between this paper and [1] from the motivation and method design perspective?

   [1] Dataset Distillation via the Wasserstein Metric (Haoyang Liu et al., 2023)
2. How is the Sinkhorn iteration $T$ decided?
3. Can the authors provide empirical evidence that introducing more models would lead to more diverse soft labels?
4. Some extra questions for experiments:
   - The authors adopt several different training settings (300 epochs, 500 epochs, and 1,000 epochs). Are the same group of soft labels incorporated for all these settings?
   - In Table 7, does the column "w/o OTG" corresponds to DiT-IGD?

**Ethical Concerns:**

["NO or VERY MINOR ethics concerns only"]

**Final Justification:**

The authors have addressed my concerns. I would like to maintain the original score of accept.

**Limitations:**

The authors have included a limitation discussion and proposed possible solutions to tackle the limitations.

**Quality:**

3

**Strengths And Weaknesses:**

## Strengths
1. The method design is based on reasonable motivations: the instance-level characteristics and intraclass variations are not taken into consideration at the same time by the previous dataset distillation methods.
2. The authors apply a detailed analysis of the decomposition of OT distance between the synthetic set and the original set. Subsequently, the authors propose three components to tackle them, respectively.
3. Extensive experiments are conducted to demonstrate the effectiveness of the proposed method. I'm surprised to see that the adopted optimal transport computation only brings negligible extra computation to the distillation process.
4. The writing is overall clear and easy to understand.

## Weaknesses
1. There was another paper utilizing optimal transport in the context of dataset distillation [1]. I can see the difference between this paper and [1]. It would further enhance the paper quality to provide a comparison and discussion.

   [1] Dataset Distillation via the Wasserstein Metric (Haoyang Liu et al., 2023)
2. It takes $T$ Sinkhorn iterations to obtain the optimal transport matrix. However, there lack of a parameter analysis of $T$. Is it designed based on empirical results or previous literature?
3. The soft labeling is a bit confusing. With a small number of teacher models (e.g., two models in this paper for IPC=10), when the models have conflicts against each other, won't that introduce more uncertainty? When more models are introduced, the voting mechanism will weaken the different opinions of each model. The authors claim that more views will introduce greater semantic diversity. It would be better to include certain empirical results to support this.

---

> ### Author Rebuttal · Authors · 2025-07-28
>
> We thank the reviewer for the constructive comments. Our responses are as follows.
>
> >	Q1: There was another paper utilizing optimal transport in the context of dataset distillation [1]. I can see the difference between this paper and [1]. It would further enhance the paper quality to provide a comparison and discussion.
>
> A1: We thank the reviewer for pointing out the related work and agree that including a comparison will further improve the paper. Although both our method and WMDD utilize optimal transport (OT), they differ significantly in both methodology and motivation, leading to distinct formulations and implementations.
>
> Liu et al. ("Dataset Distillation via the Wasserstein Metric," ICCV 2025, hereafter WMDD) is a **distribution-matching-based** distillation method that applies OT in a **single, offline step** to compute a Wasserstein barycenter over the real data’s feature distribution. This barycenter is then used as a fixed target throughout training, where synthetic images are optimized to match it using a **standard L2 loss** in the feature space. In contrast, we introduce a fundamentally different generative paradigm where OT is not a static, one-off computation, but a **dynamic guidance mechanism integrated throughout the entire data synthesis and training pipeline**. Specifically, OT guides the sampling of synthetic images by aligning latent representations, regulates the soft label relabeling process by matching label complexity to the image distribution, and structures the training loss of the student model by aligning its logits to the relabeled targets.
>
> The motivation behind WMDD is to replace the use of simple data summaries, such as the feature means often targeted by MMD-based methods, with a more geometrically meaningful summary, namely the Wasserstein barycenter, derived from the Wasserstein metric. In contrast, our method is driven by the need to address inherent limitations in generative distillation pipelines, which often fail to preserve the fine-grained geometry of the real data distribution—particularly intra-class variations and local modes. These aspects are explicitly addressed in our framework through a multi-stage, OT-guided design.
>
> We compare the top-1 accuracy of our method with WMDD under different images-per-class (IPC) settings on both the ImageNette and ImageNet-1K datasets. As shown in the table below, our method consistently outperforms WMDD across all scenarios.
>
> | Dataset | ImageNette | ImageNette | ImageNet-1K | ImageNet-1K |
> |-|-|-|-|-|
> |IPC| 10| 50| 10 | 50 |
> |WMDD| 64.8$\pm$0.4| 83.5$\pm$0.3| 38.2$\pm$0.2| 57.6$\pm$0.5|
> |Ours|**79.0$\pm$0.3**|**89.3$\pm$0.3**|**52.9$\pm$0.1**|**61.9$\pm$0.5**|
>
> >	Q2: It takes $T$ Sinkhorn iterations to obtain the optimal transport matrix. However, there lack of a parameter analysis of $T$. Is it designed based on empirical results or previous literature?
>
> A2: Thank you for the insightful question. We have conducted a targeted analysis of the number of Sinkhorn iterations $T$, as it directly influences the quality of the approximated transport matrix and the overall effectiveness of our OT-based alignment components.
>
> We summarize the results of this analysis in the table below:
>
> Table. Effect of $T$ on ImageNette (IPC=10)
> |$T$|5|10|20|50|100|
> |-|-|-|-|-|-|
> |ResNet-18|77.9|78.6|79.0|78.8|78.9|
> |ConvNet-6|73.8|74.3|74.5|74.7|74.6|
>
> As shown, increasing $T$ from 5 to 20 results in notable performance gains, particularly due to improved convergence of the OT plan. However, beyond $T$=20, the improvement plateaus, with accuracy gains becoming marginal while unnecessarily increase computational costs.  We therefore use $T$=20 by default for all reported experiments, balancing convergence quality and computational efficiency.
>
> >	Q3: The soft labeling is a bit confusing. With a small number of teacher models (e.g., two models in this paper for IPC=10), when the models have conflicts against each other, won't that introduce more uncertainty? When more models are introduced, the voting mechanism will weaken the different opinions of each model. The authors claim that more views will introduce greater semantic diversity. It would be better to include certain empirical results to support this.
>
> A3: We thank the reviewer for raising this thoughtful question. In fact, our method does not aim to reduce uncertainty by suppressing disagreement across teachers, nor do we rely on majority voting. Instead, we adopt a principled approach based on aligning the **distributional complexity** of soft labels with the **representational capacity** of the synthetic images.
>
> Concretely, using fewer teacher models results in soft label distributions that are less complex and contain fewer semantic modes. Under **low IPC** settings, where synthetic images are limited in diversity and expressiveness, a simpler soft label distribution is more suitable. It avoids over-regularizing weak representations with overly complex supervision and leads to better alignment in the joint image-label space.
>
> Conversely, under **high IPC** settings, where the synthetic image distribution is richer and contains more semantic structure, we intentionally introduce more teacher models to increase the complexity of the soft label distribution. This promotes stronger semantic supervision and better fits the higher-capacity image representations.
>
> Our Label-Image-Aligned Soft Label Relabeling (LIA) module adaptively selects a teacher ensemble based on the IPC setting. This ensures that the **complexity of the soft-label distribution matches the expressive capacity of the distilled images, thereby directly reducing the contraction factor  $\alpha$ in Eq. (6). A smaller $\alpha$ leads to a tighter OT upper bound in Eq. (8), which in turn yields better student model performance**.
>
> This adaptive strategy is not just a heuristic; it is directly and empirically validated by our analysis of $\alpha$ in Appendix G.2 (page 25):
>
> -	For Low IPC (Table 17, IPC=10): A smaller ensemble of 2 teachers (Config D) achieves a lower contraction factor $\alpha$ (0.93) and the highest accuracy (52.9% on ResNet-18). Increasing the ensemble to 4 diverse teachers (Config B) is detrimental, worsening both $\alpha$ to 0.99 and accuracy to 52.3%. This confirms that for low-capacity image sets, excessive label complexity is harmful.
>
> -	For High IPC (Table 18, IPC=50): The trend reverses as the image set's capacity increases. The larger, more diverse ensemble of 4 teachers (Config B) now achieves a dramatically smaller $\alpha$ (0.16) and the best generalization performance (68.2% on Swin Transformer). The simpler 2-teacher ensemble is now suboptimal.
>
> Together, these results validate our design: **matching soft-label distribution complexity to image capacity minimizes $\alpha$, reduces OT distance, and improves performance**. We will clarify this mechanism and include illustrative diagrams (including a motivation diagram and a causal flow) in the final version.
>
> **Additional diversity check.** To qualitatively assess whether a larger teacher ensemble introduces greater semantic diversity, we report the average Shannon entropy of the soft labels (ImageNette, IPC = 10):
> |Ensemble|Entropy|$\alpha$|
> |-|-|-|
> |4 teachers|1.91 nats|0.903|
> |2 teachers|1.82 nats|0.643|
>
> The results suggest the **presence of increased semantic diversity and richness when using more teacher models**, observed at the individual sample level. However, any single‑sample statistic cannot enumerate mode count, reveal inter‑sample structure, or fully describe the geometry of the label distribution. **It is therefore unsuitable as an optimization proxy**. In practice, we recommend: (1)  selecting a small set of ensemble configurations based on IPC, following our LIA guideline; and (2) choosing the one with the smallest $\alpha$, which directly tightens the OT upper bound (Eq. 8).
>
> >	Q4: The authors adopt several different training settings (300 epochs, 500 epochs, and 1,000 epochs). Are the same group of soft labels incorporated for all these settings?
>
> A4: No, the effective soft labels are different for each training epoch.
> To ensure a **fair comparison with state-of-the-art baselines**	(e.g., SRe2L, RDED, G-VBSM, CDA, SC-DD, CV-DD, EDC, D$^4$M, TDSDM, Minimax, DDPS, DiT-IGD), we follow the methodology similar to FKD [1]. This approach utilizes a unique set of soft labels for each training epoch, which are generated based on different stochastic data augmentations. This provides the non-repetitive supervision necessary for effective long-term training.
>
> [1] A fast knowledge distillation framework for visual recognition. ECCV, 2022
>
>
> >	Q5: In Table 7, does the column "w/o OTG" corresponds to DiT-IGD?
>
> A5: For hard label settings: Yes, the “w/o OTG” column corresponds to DiT-IGD, which uses guided diffusion without OT guidance.
>
> For soft label settings: No, the correspondence does not hold. DiT-IGD lacks all the OT components proposed by our method (OTG, LIA, OTM).

---

> > ### Comment · Reviewer_Xuuh · 2025-08-03
> >
> > I would like to thank the authors for providing the detailed response. I will keep my initial score of accept. Please do incorporate these discussions in the revised manuscript.

---

> > > ### Author Response · Authors · 2025-08-03
> > > **Thanks for your comments**
> > >
> > > We sincerely thank Reviewer Xuuh for the positive evaluation and constructive engagement. We will incorporate the discussed points into the revised manuscript to ensure clarity and completeness. Wishing you a wonderful day!

---

### Official Review · Reviewer_i9LC · 2025-07-01

**Clarity:** 2
**Significance:** 2
**Originality:** 2
**Rating:** 5
**Confidence:** 3

**Summary:**

This paper proposes a dataset distillation method based on optimal transport  to align the distribution between real and synthetic data. It introduces a three-stage sampling framework: an OT-guided sampling, a soft label relabeling, and an OT-based logit matching, to preserve both global and fine-grained structure. Experiments on ImageNet and other benchmarks show strong performance, especially in low-data settings.

**Questions:**

1. The notation is confusing. For example, writing “S ≡ ν_distill(x, y)” and “T ≡ μ_true(x, y)” mixes up datasets (which are collections of points) with distributions (which are probabilistic objects). Later, in Eq. (5), both terms use ν_distill, but one is about image data and the other is about label logits. These are very different things, so reusing the same symbol makes it hard to follow the logic.

2. It’s unclear how Eq. (8) is derived from Eq. (7). Why the inequality hold?

**Ethical Concerns:**

["NO or VERY MINOR ethics concerns only"]

**Final Justification:**

As they have addressed most of my concerns, I will increase my rating to accept.

**Limitations:**

1. The paper feels a bit over-sold. The idea of distribution matching is in a heavy optimal transport fashion. The optimal transport are in three parts of the pipeline, but only the sampling step clearly needs it. The other two (label relabeling and logit matching) could probably be done with simpler methods like MMD or KL divergence. Also, the idea of using OT for dataset distillation isn’t new — see:
Liu et al., "Dataset Distillation via the Wasserstein Metric", arXiv:2311.18531 (2023).

2. The results look good, but they need long training to get there. For example, the best results on ImageNet-1K require 1000 training epochs, which weakens the claim that this is an efficient method. The paper should discuss performance under shorter training more directly.

3. Small IPC cases (<10) are not well covered. IPC=10 is already a relatively generous setting. Many real-world use cases care about much lower IPC, like 1 to 5. It would be helpful to see how the method performs in these tougher scenarios.

**Quality:**

3

**Strengths And Weaknesses:**

Strengths
The performances is good. The model outperforms baselines across datasets and architectures, especially at IPC=10.

Weaknesses
1. The paper is a bit over-selling on simple idea. OT is used in all stages, though simpler methods may work for some. and OT idea in dataset distillation is not new.
2. The notation is ambiguous, the same symbol is  used for both data and label distributions, causing ambiguity.

---

> ### Author Rebuttal · Authors · 2025-07-28
>
> We thank the reviewer for the constructive comments. Our responses are as follows.
>
> >	Q1: The notation is confusing. For example, writing “S ≡ ν_distill(x, y)” and “T ≡ μ_true(x, y)” mixes up datasets (which are collections of points) with distributions (which are probabilistic objects). Later, in Eq. (5), both terms use ν_distill, but one is about image data and the other is about label logits. These are very different things, so reusing the same symbol makes it hard to follow the logic.
>
> A1: First, regarding writing the dataset $\mathcal{S}$ as the distribution $\nu_{\text{distill}}(\mathbf{x},\mathbf{y})$, this is indeed an academic shorthand where we treat a finite dataset as its empirical distribution [1].  To avoid ambiguity, we will clarify this convention explicitly in a footnote in the final version (e.g., “$\nu_{\text{distill}}$ denotes the empirical distribution over dataset $\mathcal{S}$”).
>
> Second, regarding the seeming reuse of symbols for "image data" and "label logits" in Equation (5), we want to clarify that, as explicitly defined in Table 1, both Wasserstein distance terms operate over the full joint distribution $(\mathbf{x},\mathbf{y})$. That is, the notation $\nu_{\text{distill}}$ consistently denotes the same object: the distilled joint data-label distribution. Our framework's two-stage "relay race" design helps to understand this:
>
> -	Stage 1: $\mathrm{W}(\mu_{\text{true}}, \nu_{\text{distill}}^{(\mathrm{soft})})$ measures the gap between the real data and our distilled data. It asks, "How well do our synthetic (image, soft label) pairs represent the true distribution?" This is optimized by our image generation (OTG) and label assignment (LIA) modules.
>
> -	Stage 2: $\mathrm{W}(\nu_{\text{distill}}^{(\mathrm{soft})}, \nu_{\text{new}})$ measures the gap between our distilled data and the final student model's output. The logits are merely the mechanism for optimizing this stage, while the objective remains matching the joint distributions.
>
> To formally eliminate this ambiguity at the symbolic level throughout the paper, we will explicitly write (x,y) in all relevant distribution notations in our revision, for example, by writing $\mathrm{W}(\mu_{\text{true}}, \nu_{\text{new}})$ as $\mathrm{W}(\mu_{\text{true}}(\mathbf{x},\mathbf{y}), \nu_{\text{new}}(\mathbf{x},\mathbf{y}))$.
>
> [1] G. Peyré and M. Cuturi. Computational Optimal Transport: With Applications to Data Science. Foundations and Trends in Machine Learning, 2019.
>
> >	Q2: It’s unclear how Eq. (8) is derived from Eq. (7). Why the inequality hold?
>
> A2: Equation (8) is formed by substituting Equations (6) and (7) back into Equation (5):
>
> - First, replace $ \mathrm{W}(\mu_{\text{true}}, \nu^{(\mathrm{soft})}_{\text{distill}}) $ in Equation (5) with its equivalent from Equation (6).
> - Then, replace $ \mathrm{W}(\mu_{\text{true}}, \nu^{(\mathrm{hard})}_{\text{distill}}) $ with its class-wise expectation from Equation (7).
>
> This leads to the final structured upper bound in Equation (8).
>
> >	Q3: The paper feels a bit over-sold. The idea of distribution matching is in a heavy optimal transport fashion. The optimal transport are in three parts of the pipeline, but only the sampling step clearly needs it. The other two (label relabeling and logit matching) could probably be done with simpler methods like MMD or KL divergence. Also, the idea of using OT for dataset distillation isn’t new — see: Liu et al., "Dataset Distillation via the Wasserstein Metric", arXiv:2311.18531 (2023).
>
> A3: Thank you for your insightful feedback. Our contribution is not in merely using OT, but in structurally integrating OT across sampling, relabeling, and logit‑matching as a single end‑to‑end OT distance minimization objective, which is essential for preserving distributional geometry. Results show the necessity of each component and the superiority of our whole pipeline.
>
> On the Logit Matching component, KL and MMD indeed serve as simpler divergences, but they are inherently limited. KL divergence is applied per sample and ignores inter-sample relationships, while MMD matches only global statistics. In contrast, our OTM applies **batch-wise OT alignment** between student logits and soft labels, capturing the **joint distributional structure** of samples. This enables OT to faithfully preserve **inter-sample geometry** and match structural uncertainty in the soft labels, which KL and MMD overlook. As shown below, this leads to a clear performance gain:
>
> Table. Performance comparison for logit matching on ImageNette (IPC=10).
>
> |Network|ConvNet-6| ResNetAP-10|ResNet-18|
> |-|-|-|-|
> |MMD|72.6|75.3|76.4|
> |KL|73.0|75.4|77.6|
> |Ours (OTM)|**74.5**|**77.8**|**79.0**|
>
> The LIA module serves as a **contraction-control strategy** grounded in our OT framework, and its effectiveness is validated by extensive experiments in Tables 10, 17, and 18. It adaptively selects teacher ensembles based on the IPC setting and controls the **soft label distribution complexity** to match image expressiveness. This design directly **minimizes the OT-derived contraction factor $\alpha$**, which tightens the OT upper bound (Eq. 8), thereby enhancing the effectiveness of supervision. In contrast, KL and MMD are not only less expressive but are also incompatible with our OT-based framework. They **cannot guide teacher ensemble selection nor participate in contraction-aware optimization**, which are essential to our LIA module.
> Therefore, **neither LIA nor OTM can be substituted by KL or MMD.**
>
> Regarding Liu et al. ("Dataset Distillation via the Wasserstein Metric," ICCV 2025, hereafter WMDD), while both WMDD and our approach leverage OT, they differ fundamentally in both scope and formulation. WMDD is designed to for **distribution-matching-based dataset distillation** and applies OT only once at the beginning of training to compute a static Wasserstein barycenter. This barycenter is then matched using a conventional L2 loss.
> In contrast, our method introduces a holistic framework for **generative dataset distillation**. Our method employs dynamic OT guidance in three stages—sampling, relabeling, and logit matching—forming a unified, multi‑stage pipeline that optimizes the same OT objective end to end. Also, WMDD uses OT for **offline initial computation**, whereas we employ OT **actively throughout training** to preserve evolving distributional geometry.
> This distinction results in a consistent performance gap, as shown in the table below:
>
> |Dataset|ImageNette|ImageNette| ImageNet-1k|ImageNet-1k|
> |-|-|-|-|-|
> |IPC|10|50|10|50|
> |WMDD|64.8$\pm$0.4|83.5$\pm$0.3|38.2$\pm$0.2|57.6$\pm$0.5|
> |Ours|**79.0$\pm$0.3**|**89.3$\pm$0.3**|**52.9$\pm$0.1**|**61.9$\pm$0.5**|
>
> We will include a discussion of WMDD in the related work section and incorporate it as a new baseline in the experimental evaluation in the revised manuscript.
>
> >	Q4: The results look good, but they need long training to get there. For example, the best results on ImageNet-1K require 1000 training epochs, which weakens the claim that this is an efficient method. The paper should discuss performance under shorter training more directly.
>
> A4: Thank you for raising this important point regarding efficiency. We would like to clarify that the 1000-epoch results were presented not as a requirement of our method, but as part of the **post-distillation evaluation** to demonstrate the information richness and long-term utility of the distilled dataset. However, such **training of the downstream model is not necessary** to achieve strong performance.
> As detailed in Table 2, when training a new ResNet-18 for only 300 epochs, our method achieves 52.9% accuracy on ImageNet-1K (IPC=10). This result significantly outperforms leading baselines under the same settings, such as the top model-inversion method EDC (48.6%) and the top generative method DiT-IGD (45.5%).
>
> Furthermore, the overall efficiency of our method is highlighted by its **rapid convergence during downstream training** and **low-cost distilled data generation**, which is a **central goal in dataset distillation**. The **new model training curves in Figure 3** show that our method "achieves faster convergence and consistently higher accuracy, especially in early epochs" when compared to EDC and IGD. This demonstrates the high quality and informativeness of our distilled samples. In addition to training dynamics, the **generation of the distilled set itself is also highly efficient**. Table 8 shows that the adopted OT computation only brings **negligible extra computation** to the distillation process. Our runtime analysis in Table 9 reveals that our approach is notably **faster than EDC**.
>
> >	Q5: Small IPC cases (<10) are not well covered. IPC=10 is already a relatively generous setting. Many real-world use cases care about much lower IPC, like 1 to 5. It would be helpful to see how the method performs in these tougher scenarios.
>
> A5: We thank the reviewer for this insightful suggestion. We have conducted additional experiments on ImageNet-1K for the challenging settings of IPC=1, IPC=2, and IPC=5. The results of these new experiments are presented in the three tables provided below.
>
> Importantly, in the IPC = 1 setting, since only one synthetic image is generated per class, the OTG process cannot leverage previously distilled samples for alignment. Instead, for each class, we compute the OT distance between its single synthetic candidate and the corresponding real images in the latent space to guide generation.
>
> Table: Performance on ImageNet-1K (IPC=1)
> |Method|DM|FrePo|TESLA|SRe2L|RDED|EDC|DiT-IGD|Ours|
> |-|-|-|-|-|-|-|-|-|
> |Acc|1.5|7.5|7.7|0.4|6.6|12.8|10.7|**15.9**|
>
> Table: Performance on ImageNet-1K (IPC=2)
> |Method|DM|FrePo|TESLA|RDED|EDC|DiT-IGD|Ours|
> |-|-|-|-|-|-|-|-|
> |Acc|1.7|9.7|10.5|16.5|22.8|20.6|**25.9**|
>
> Table: Performance on ImageNet-1K (IPC=5)
> |Method|RDED|EDC|DiT-IGD|Ours|
> |-|-|-|-|-|
> |Acc|23.8|39.5|38.6|**45.7**|
>
> Our results consistently surpass prior state-of-the-art methods.

---

> > ### Comment · Reviewer_i9LC · 2025-08-04
> >
> > Thanks to the authors for their efforts during the rebuttal. As they have addressed most of my concerns, I will increase my rating accordingly.

---

> > > ### Author Response · Authors · 2025-08-04
> > > **Thanks for your comments**
> > >
> > > Thank you for your valuable feedback. Best wishes to you!

---

### Official Review · Reviewer_kJL2 · 2025-07-02

**Clarity:** 3
**Significance:** 2
**Originality:** 3
**Rating:** 4
**Confidence:** 4

**Summary:**

This paper observes that recent methods for large-scale dataset distillation primarily focus on matching global distributional statistics (e.g., mean and variance) while overlooking critical instance-level characteristics and intra-class variation, which limits generalization. To address this, the authors reformulate dataset distillation as an Optimal Transport distance minimization problem, enabling fine-grained alignment at both global and instance levels. OT provides a geometric framework for distribution alignment, preserving local modes, intra-class structure, and high-dimensional distributional geometry. Their method comprises three components designed to preserve distributional structure: 1) OT-guided diffusion sampling to align the latent distributions of real and distilled images. 2) Label-image-aligned soft relabeling, which adjusts label distributions according to the complexity of distilled image distributions. 3) OT-based logit matching to align student model outputs with the soft-label distribution. Experiments across multiple architectures and datasets show that the proposed approach outperforms existing methods, achieving ~4% accuracy gain under the IPC=10 setting for each architecture on ImageNet-1K.

**Questions:**

Please see the *Strengths And Weaknesses* part above. My primary concern is whether OT provides a meaningful advantage for this task, and why it is truly needed. I would prefer not to see it used merely for novelty (e.g., just because no prior DD papers have used it) without concrete justification or practical benefit. In addition, some implementation details are missing, and I hope the authors will address these concerns thoroughly in the rebuttal.

**Ethical Concerns:**

["NO or VERY MINOR ethics concerns only"]

**Final Justification:**

The authors' response has generally addressed my concerns. My initial major doubt was whether combining Optimal Transport (OT) with dataset distillation was truly necessary. The authors explained that the benefit lies in capturing critical instance-level features and fine-grained geometric structures, which I find to be a reasonable justification.

**Limitations:**

The authors have discussed some of their method's limitations in the paper.

**Paper Formatting Concerns:**

No.

**Quality:**

3

**Strengths And Weaknesses:**

Strengths
1. Applying Optimal Transport to generative dataset distillation appears novel as I am not aware of prior work using it in this context.
2. The paper is generally well-written.
3. The idea of aligning both global and instance-level distributions in the distillation process is interesting.
4. The authors have included their code in the submission, which enhances the reliability of their results and shows good improvements over previous methods.


Weaknesses
1. The paper misses some technical details. For example, in Section 4.4 (label-image-aligned soft relabeling), the authors do not clarify whether soft labels are generated at the global image level or like SRe2L at the region level. These different operations could lead to significant differences in performance.
2. In Tables 2 and 3, results from different training epochs (300, 500, 100) are presented in the same table. It would be more intuitive to split these into separate tables to allow for direct comparison with other baselines trained for the same number of epochs.
3. The motivation for using Optimal Transport seems reasonable, I am concerned this may be another instance of a "forced A+B" combination, where a well-known technique is applied simply for novelty rather than clear necessity or demonstrated advantage.
4. The overall pipeline appears relatively complex, especially the OT-guided diffusion sampling part, which may reduce its utilization in practice.

---

> ### Author Rebuttal · Authors · 2025-07-28
>
> We thank the reviewer for the constructive comments. Our responses are as follows.
>
> >	Q1: The paper misses some technical details. For example, in Section 4.4 (label-image-aligned soft relabeling), the authors do not clarify whether soft labels are generated at the global image level or like SRe2L at the region level. These different operations could lead to significant differences in performance.
>
> A1：For a **fair comparison** with prior methods (e.g., SRe2L, RDED, G-VBSM, CDA, SC-DD, CV-DD, EDC, D$^4$M, TDSDM, Minimax, DDPS, DiT-IGD), we adopt the same region-level soft label storage strategy as in FKD [1]. Notably, **except for D$^3$M, all baselines in Tables 2 and 3 follow this convention**. We will explicitly clarify this detail in Section 4.4 of the final manuscript, to avoid confusion and ensure technical transparency.
>
> [1] A fast knowledge distillation framework for visual recognition. ECCV, 2022
>
> >	Q2：In Tables 2 and 3, results from different training epochs (300, 500, 1000) are presented in the same table. It would be more intuitive to split these into separate tables to allow for direct comparison with other baselines trained for the same number of epochs.
>
> A2: Thank you for highlighting the clarity issue in Tables 2 and 3.
>
> We agree that facilitating direct comparison is important. Our primary motivation for presenting the 300-, 500-, and 1000-epoch results together is to demonstrate that the distilled data is sufficiently rich to support continued performance improvement with longer training. The 300-epoch results are included for a direct and fair comparison, as 300 epochs represent the most commonly adopted (and typically the shortest) training schedule in prior work on ImageNet-1K. However, as you note, this standard is not universal; for instance, Minimax and DDPS use 2000 epochs for IPC=10, 1500 for IPC=50, and 1000 for IPC=100, while SC-DD uses 600 epochs for each IPC.
>
> To address your concern while transparently presenting the full scope of our findings, we will revise the table layout in our final manuscript. Specifically, we will adopt the following structure:
>
> 1. We will add a dedicated "Epochs" column to clearly and unambiguously state the training epochs for every method.
>
> 2. We will apply a light‐gray background to all 300‑epoch entries so readers can identify the primary, apples‑to‑apples comparison at a glance.
>
> 3. We will revise the table captions to explicitly state that the 300-epoch results serve as the primary point of comparison, while the 500- and 1000-epoch results demonstrate the continued improvement of our method.
>
> >	Q3: The motivation for using Optimal Transport seems reasonable, I am concerned this may be another instance of a "forced A+B" combination, where a well-known technique is applied simply for novelty rather than clear necessity or demonstrated advantage.
>
> A3: We thank the reviewer for this critical question, as it touches upon the core of genuine research innovation. We wish to clarify that our framework is not a "forced combination" of Optimal Transport (OT) and dataset distillation for novelty's sake. Instead, our use of OT is a principled response to a fundamental limitation we identified in prior state-of-the-art methods. Existing advanced approaches in large-scale dataset distillation primarily focus on matching global statistics, such as mean and variance. However, this approach has an inherent weakness: **it overlooks the critical instance-level features and fine-grained geometric structures** within the data distribution, such as the various local modes or intra-class variations. By overlooking geometric details, the resulting distilled datasets fail to capture the real data's underlying structure and diversity, which in turn limits the generalization capability of models trained on such data. To overcome this bottleneck, a new theoretical framework capable of fundamentally understanding and preserving distributional geometry is needed, and Optimal Transport is the ideal choice for this task.
>
> Therefore, Optimal Transport is not an ancillary module in our method but its foundational principle throughout. We **reformulate the entire dataset distillation task as an end-to-end OT distance minimization problem**. This objective is then ingeniously decomposed into three synergistic stages of the distillation pipeline, ensuring geometric distribution alignment from start to finish. First, in the image generation stage, we employ "OT-guided diffusion sampling" to align the latent space distributions of real and generated images, thereby preserving complex local patterns. Second, in the label assignment stage, our "label-image-aligned soft relabeling" strategy adapts the complexity of the label distribution to match that of the distilled images, which is directly tied to minimizing the overall OT distance via a contraction factor. Finally, during student model training, we use "OT-based logit matching," which aligns model outputs with soft labels at the batch level, effectively preserving the relational structure between samples that per-sample metrics would ignore.
>
> The necessity and superiority of this OT-centric approach are not merely theoretical but are robustly and clearly demonstrated in our experiments. Most critically, our ablation studies in Table 7 provide direct evidence against the "forced combination" concern. The results show that when **removing any of the proposed OT-guided components (OTG, LIA, or OTM), the performance of all tested architectures drops significantly**, proving that these modules are indispensable for achieving our reported performance. This superior alignment translates into substantial performance gains, where our method consistently outperforms prior art, **achieving at least a 4% accuracy improvement on ImageNet-1K under the IPC=10 setting across all tested architectures**. Furthermore, the visualizations in Figure 2 confirm that our method produces **more diverse and representative samples that closely approximate the real data manifold**, whereas prior approaches often yield perceptibly blurred or semantically inaccurate generations.
>
> We also demonstrate that **our individual OT-based modules are superior to common, simpler alternatives for their specific tasks**. For the sampling stage, Table 19 shows our **OT-Guided Sampling (OTG) significantly outperforms guidance based on MMD (Euclidean) and MMD (RKHS)**. This performance gap arises because metrics like MMD matches only aggregate moments and fails to explicitly model instance-level correspondences. In contrast, OT establishes pairwise couplings between real and synthetic samples, **enabling finer-grained alignment that better preserves the geometric structure of the data manifold**. For the relabeling stage, Tables 10, 17, and 18 demonstrate that our **Label-Image-Aligned (LIA) relabeling module consistently yields lower values of the contraction factor $\alpha$** compared to randomly or statically selected teacher ensembles. This **reduction in $\alpha$ directly translates into substantial performance improvements**. For the logit matching stage, the following results on ImageNette (IPC=10) show that our **OT-based Matching (OTM) consistently outperforms both KL Divergence and MMD**:
>
> Table. Performance comparison of logit matching methods on ImageNette (IPC=10).
>
> |Network|ConvNet-6| ResNetAP-10|ResNet-18|
> |-|-|-|-|
> |MMD|72.6|75.3|76.4|
> |KL|73.0|75.4|77.6|
> |Ours (OTM)|**74.5**|**77.8**|**79.0**|
>
> Per-sample methods like KL divergence are applied individually to each sample and ignore inter-sample relationships, while global metrics like MMD match only global statistics. In contrast, our **batch-wise OT (Sinkhorn) alignment** (OTM) captures the **joint distributional structure** of the logits and faithfully preserves **inter-sample geometry**, which other methods overlook.
>
> In conclusion, we demonstrate that OT is a principled and effective method for resolving the geometric deficiencies of existing distillation methods, with its necessity and advantages fully validated by our experimental results.
>
> >	Q4: The overall pipeline appears relatively complex, especially the OT-guided diffusion sampling part, which may reduce its utilization in practice.
>
> A4: We thank the reviewer for their valuable feedback regarding the perceived complexity of our pipeline. While the pipeline includes multiple stages, we demonstrate that it remains lightweight and straightforward to use. In fact, the OT-guided diffusion sampling part is implemented with minimal overhead. **Runtime profiling in Table 8 shows < 1 % extra cost.** Furthermore, the complete generation pipeline is notably faster than competing state-of-the-art methods such as EDC, with increasing advantage as the number of GPUs increases, highlighting its practicality for large-scale applications.
>
> Beyond its computational efficiency, we designed the framework for straightforward adoption. A notable strength of our method is that **a fixed set of OT-related hyperparameters generalizes well across datasets and model architectures**. Moreover, performance remains robust even under moderate parameter variations, as shown in Figures 4–7, demonstrating **low sensitivity and eliminating the need for exhaustive tuning**. This robustness significantly reduces the burden of deployment in new scenarios.
>
> The necessity of each component, including the OT-guided sampling, is empirically validated by our ablation study, where removing any stage results in a significant degradation of performance. This confirms that the pipeline's structure is not unnecessarily complex but rather essential for preserving the fine-grained distributional geometry that enables our method to significantly outperform prior work. We will ensure these points on efficiency, ease of use, and justified design are more prominently highlighted in the final manuscript to better convey the practical value of our approach.

---

> > ### Comment · Reviewer_kJL2 · 2025-08-05
> >
> > Thanks for the authors' responses, they basically addressed most of my questions.

---

> > > ### Author Response · Authors · 2025-08-05
> > > **Thanks for your comments**
> > >
> > > We sincerely thank the reviewer for the positive feedback. We are glad that our responses addressed the raised concerns. We will incorporate the relevant clarifications into the final version of the manuscript.

---

### Official Review · Reviewer_y7XW · 2025-07-03

**Clarity:** 3
**Significance:** 3
**Originality:** 3
**Rating:** 4
**Confidence:** 2

**Summary:**

This paper proposes a novel dataset distillation method based on Optimal Transport (OT) aimed at achieving geometrical alignment between real and distilled data by minimizing OT distance. The core idea of the approach is to preserve local modes, intra-class patterns, and fine-grained variations in high-dimensional distributions, thereby enhancing the quality and generalization ability of the distilled data.

**Questions:**

Q1.  In Experiment Table 11, I noticed that the values of \( \lambda_1 \) are set to 1000 and 10000, which seems unreasonable. As $\lambda_1$ increases, the solution to the Sinkhorn algorithm becomes more uniform, leading to a weaker correlation with the image information. I suggest that the authors provide experiments with smaller values of \( \lambda_1 \) to alleviate our concerns about the first part of the guidance not having a significant effect.

Q2. I found that Section 4.5, "OT-based Student Model Logit Matching," is somewhat similar to a recent paper [2]. I would appreciate it if the authors could provide a comparison and highlight the differences between the two.


[2] SelKD: Selective Knowledge Distillation via Optimal Transport Perspective. ICLR2025

**Ethical Concerns:**

["NO or VERY MINOR ethics concerns only"]

**Limitations:**

Yes

**Quality:**

3

**Strengths And Weaknesses:**

Strengths:

1. The motivation in Figure 1(a) is interesting, and the use of Wasserstein Distance in this context is reasonable.

2. I appreciate the explanation in Section 4.2, "Reconstructing the Optimal Transport Distance," where the author summarizes the three methods of the paper through an approximated inequality.

Weaknesses:

1. In Eq. 13, the author refers to $W=⟨D,K⟩$ as the optimal transport distance, which is not entirely accurate. Strictly speaking, it is an approximation of the Wasserstein Distance, or more specifically, the Sinkhorn distance, since $K$ is just an approximation of the exact solution. For more details, see [1].

2. The author proposes three improvements, but I am unclear about which one plays a decisive role.

3. I do not fully understand the significance of Label-Image-Aligned Soft Label Relabeling. I would appreciate it if the author could provide some diagrams to help readers understand, instead of a long explanation in text.



[1] Sinkhorn distances: Lightspeed computation of optimal transport

---

> ### Author Rebuttal · Authors · 2025-07-28
>
> We thank the reviewer for the constructive comments. Our responses are as follows.
>
> >	Q1: In Eq. 13, the author refers to $W = \langle D, K \rangle$ as the optimal transport distance, which is not entirely accurate. Strictly speaking, it is an approximation of the Wasserstein Distance, or more specifically, the Sinkhorn distance, since $K$ is just an approximation of the exact solution.
>
> A1: We thank the reviewer for the accurate observation. We agree that $\mathrm{W} $ is the Sinkhorn distance — an entropy-regularized approximation of the true Wasserstein distance. In our initial presentation, we used the term "optimal transport distance" to maintain notational simplicity, as the Sinkhorn method is a standard and computationally necessary approximation for high-dimensional data where the exact Wasserstein distance is intractable. In the final version, we will revise our phrasing to avoid any ambiguity. Specifically, we will **replace “optimal transport distance” with “Sinkhorn distance (an approximation of the OT distance)”** when referring to Eq. 13 and related components.
>
> >	Q2: The author proposes three improvements, but I am unclear about which one plays a decisive role.
>
> A2: We sincerely thank you for this insightful question. It touches upon the core philosophy of our method's design, and we appreciate the opportunity to clarify the interplay between the three main components of our framework.
>
> We wish to first clarify that the three components—(1) OT-guided Diffusion Sampling, (2) Label-Image-Aligned Soft Relabeling, and (3) OT-based Logit Matching—are not designed as independent "improvements." Instead, they are three deeply interconnected and indispensable stages of a single, unified framework. Our core contribution is to reformulate the entire dataset distillation task as an end-to-end OT distance minimization problem, aiming to minimize $\mathrm{W}(\mu_{\text{true}},\nu_{\text{new}})$. The three components the reviewer mentioned are precisely the optimization sub-goals derived from a principled, theoretical decomposition of this main objective (as shown in Equation 8 of our paper). They work synergistically to serve the final goal of distributional geometry alignment across the entire pipeline.
>
> To empirically validate that each component plays a critical, non-redundant role, we conducted a detailed Ablation Study, with the results fully demonstrated in Table 7 (page 8). The results clearly show that:
> The full framework achieves the highest accuracy.
> Removing any single component leads to a significant performance drop.
> This experiment strongly proves that all three components are integral to our framework's success. There is no single "decisive" component; rather, the strength of our method lies in their synergistic integration, which systematically aligns distributional geometry from image generation to student model training. We believe it is this holistic design, where each part addresses a specific term of the overall OT objective, that ultimately leads to our state-of-the-art performance.
>
> We will make sure to further emphasize this synergistic relationship in the final version of our paper.
>
> >	Q3: I do not fully understand the significance of Label-Image-Aligned Soft Label Relabeling. I would appreciate it if the author could provide some diagrams to help readers understand, instead of a long explanation in text.
>
> A3: We thank the reviewer for suggesting to include diagrams. Below, we present pseudo-diagrams and empirical evidence to clarify our method.
>
> Motivation: match soft-label distribution complexity to distilled image distribution capacity
>
> Higher IPC →  More complex distilled image distribution (higher diversity and expressiveness)
>
> Larger teacher ensemble (when IPC is high)  → More complex soft label distributions with richer semantic modes
>
> Causal chain:
>
> Given an IPC setting (e.g., 10 or 50)
>
>  → LIA selects an IPC-adaptive teacher ensemble
>
>  → Soft-label distribution complexity matches synthetic image capacity
>
>  → Contraction factor $\alpha$ in Eq. (6) decreases
>
>  → OT upper bound in Eq. (8) tightens
>
>  → Top-1 accuracy improves
>
> As illustrated above, our Label-Image-Aligned Soft Label Relabeling (LIA) module adaptively selects a teacher ensemble based on the IPC setting. This ensures that the **complexity of the soft-label distribution matches the expressive capacity of the distilled images, thereby directly reducing the contraction factor  $\alpha$ in Eq. (6). A smaller $\alpha$ leads to a tighter OT upper bound in Eq. (8), which in turn yields better student model performance**.
>
> We validate both the effectiveness of LIA in reducing $\alpha$ and the downstream benefit of lower $\alpha$ on student accuracy, across small- and large-scale datasets.
>
> - **LIA ⟶ Smaller $\alpha$:**
>   - In **Table 10** (*ImageNette*), applying LIA reduces  $\alpha$ from 0.906 to 0.643 under IPC=10.
>   - In **Tables 17 and 18** (*ImageNet-1K*), teacher configurations selected via LIA also consistently yield the lowest $\alpha$ among all variants.
>
> - **Smaller $\alpha$ ⟶ Higher Accuracy:**
>   - In all three tables, the setting with the **smallest $\alpha$** also achieves the **best Top-1 accuracy**, confirming that $\alpha$ is not only reduced but consequential to performance.
>
> These results confirm that LIA reduces the OT alignment gap by controlling the contraction factor $\alpha$, thus contributing directly to the overall generalization quality of the student model. We will insert the above diagrams (with proper styling) into the appendix and cross‑reference it in Section 4.4 to enhance readability.
>
> >	Q4: In Experiment Table 11, I noticed that the values of ( \lambda_1 ) are set to 1000 and 10000, which seems unreasonable. As λ1 increases, the solution to the Sinkhorn algorithm becomes more uniform, leading to a weaker correlation with the image information. I suggest that the authors provide experiments with smaller values of ( \lambda_1 ) to alleviate our concerns about the first part of the guidance not having a significant effect.
>
> A4: We sincerely thank the reviewer for this thoughtful observation. Our choice of $\lambda_1$ is driven by two key considerations: (1) the numerical stability of the Sinkhorn kernel and (2) alignment with the scale of the latent space in our framework. Specifically, the cost matrix $\mathbf{D}$, computed in the latent space, have a **mean value typically falling in the range of 2000 to 3000**. Given that the transport kernel is computed as $\exp(-\mathbf{D} / \lambda_1)$, using a small $\lambda_1$ (e.g., 10 or 100) would result in extreme exponentials (e.g., $\exp(-300)$), causing the kernel to underflow to near-zero entries. This in turn leads to a degenerate or uninformative transport plan, effectively nullifying gradient flow and rendering the guidance ineffective.
>
> To ensure that the kernel matrix remains well-conditioned and that the resulting transport plan captures meaningful structure, $\lambda_1$ must be chosen on a scale commensurate with the cost values in $\mathbf{D}$. In our experiments, we found $\lambda_1 = 1000$ to be a stable and effective setting, balancing expressiveness with numerical tractability. This choice is empirically supported by the sensitivity analysis presented in Appendix G.1 (Figure 5, page 22), where we evaluate a wide range of $\lambda_1$ values from 100 to 10,000. As shown, performance degrades significantly for small $\lambda_1$ values, while peaking around $\lambda_1 = 1000$. Hence, our selected values are not arbitrary; they are grounded in both theoretical insight and empirical validation.
>
> We will revise the appendix to explicitly include a note on the scale of $\mathbf{D}$ and its influence on $\lambda_1$, to better clarify this design choice.
>
> >	Q5:  I found that Section 4.5, "OT-based Student Model Logit Matching," is somewhat similar to a recent paper [2]. I would appreciate it if the authors could provide a comparison and highlight the differences between the two.
>
> A5:  We thank the reviewer for pointing out the relation to SelKD. While both methods utilize optimal transport (OT), they are designed for fundamentally different tasks, leading to distinct goals and formulations.
> SelKD introduces a flexible framework for **selective knowledge distillation**, where a student model is trained to learn only a subset of the teacher’s knowledge. It employs OT to **reconstruct a class-subset distribution by projecting the full teacher distribution onto the selected classes in a transport-cost-minimizing way**. The student is then aligned to this projected distribution using **KL divergence**, rather than directly minimizing the transport cost itself. This strategy is **well-suited for selective knowledge distillation or open-set problems**. However, in **standard closed-set full-class tasks** where all classes are retained, such teacher-label reallocation is **unnecessary**, rendering the **OT component of SelKD less relevant and offering limited benefits in this setting**.
>
> In contrast, our work focuses on dataset distillation, where the goal is to synthesize a small dataset that enables new models to generalize across the **full label space**. The OT-based logit matching in Section 4.5 is one component within our broader framework. We apply the batch-wise Sinkhorn distance between student logits and generated soft labels to enhance supervision within each distilled mini-batch, effectively **capturing the distributional structure across all classes while preserving inter-sample geometry**.
>
> To provide a direct comparison, we created a variant of our framework by replacing our OT-based Logit Matching (OTM) module with the loss function proposed by SelKD for the standard closed-set task. We denote this configuration as 'Ours (w/ SelKD loss)' :
>
> Table. Performance comparison on ImageNette (IPC=10)
> | Network|ConvNet-6|ResNetAP-10|ResNet-18|
> |-|-|-|-|
> |Ours (w/ SelKD loss)|72.6|74.2|75.8|
> |Ours|**74.5**|**77.8**|**79.0**|

---

> > ### Comment · Reviewer_y7XW · 2025-08-06
> >
> > I thank the authors for providing a detailed response. I maintain my initial score due to another paper I recently read, 'WMDD: Dataset Distillation via the Wasserstein Metric,' which uses Wasserstein Barycenter to address the DD problem. Compared to directly using Sinkhorn in Section 4.5, the use of the barycenter may be more intuitive

---

> ### Author Response · Authors · 2025-08-06
> **Discussion and Comparison with WMDD**
>
> We sincerely thank the reviewer for the thoughtful feedback and for raising the comparison with the WMDD method.
>
> We acknowledge that WMDD ("Dataset Distillation via the Wasserstein Metric", ICCV 2025) presents an elegant solution by using the Wasserstein barycenter as a fixed distributional target for dataset distillation. However, while both our method and WMDD utilize Optimal Transport (OT), they are **fundamentally different in motivation, formulation, and implementation**.
> WMDD adopts a **distribution-matching-based** paradigm that computes the Wasserstein barycenter of the real feature distribution in a **single, offline step**. This barycenter then serves as a **static target** throughout training, and synthetic images are optimized to match it via a **standard L2 loss** in feature space. While intuitive, this approach treats OT as a one-time summarization and lacks adaptability during the training dynamics.
>
> In contrast, we introduce a fundamentally different generative paradigm where OT is not a static, one-off computation, but a **dynamic guidance mechanism integrated throughout the entire data synthesis and training pipeline**:
>
> -	**Data Sampling**: OT is used to guide the latent alignment of synthetic images with real data, improving the diversity and representativeness of generated samples.
> -	**Soft Label Relabeling**: We match the complexity of soft labels with the expressive capacity of the synthetic data, thereby tightening the OT bound, which in turn leads to better student model performance.
> -	**Student Training**: OT structures the loss by aligning student logits with relabeled targets using Sinkhorn divergence, preserving fine-grained semantic relations.
>
> The motivation behind WMDD is to replace the use of simple data summaries, such as the feature means often targeted by MMD-based methods, with a more geometrically meaningful summary, namely the Wasserstein barycenter, derived from the Wasserstein metric. In contrast, our method is driven by the need to address **inherent limitations in generative distillation pipelines**, which often fail to preserve the **fine-grained geometry of the real data distribution—particularly intra-class variations and local modes**. These aspects are explicitly addressed in our framework through a multi-stage, OT-guided design.
>
> To empirically validate the superiority of our method, we compare its performance with WMDD under various IPC (images per class) settings on both ImageNette and ImageNet-1K. As shown below, our method **consistently achieves significantly higher top-1 accuracy**:
>
> |Dataset|ImageNette|ImageNette| ImageNet-1k|ImageNet-1k|
> |-|-|-|-|-|
> |IPC|10|50|10|50|
> |WMDD|64.8$\pm$0.4|83.5$\pm$0.3|38.2$\pm$0.2|57.6$\pm$0.5|
> |Ours|**79.0$\pm$0.3**|**89.3$\pm$0.3**|**52.9$\pm$0.1**|**61.9$\pm$0.5**|
>
> We will include a discussion of WMDD in the related work section and incorporate it as a new baseline in the experimental evaluation in the revised manuscript.
>
> *The reviewer notes that WMDD's barycenter approach seems more intuitive than our use of Sinkhorn in **Section 4.5**. We would like to respectfully clarify that these two components **serve fundamentally different purposes in their respective pipelines**. WMDD's barycenter is a core part of its **initial data synthesis stage**. In contrast, our OT-based student model logit matching (Section 4.5) is a novel loss function for **the final stage of training a student model on our distilled data, a stage WMDD does not focus its OT contributions on**.*

---

### Note · Authors · 2025-08-12

Dear reviewers, ACs, and SACs,

We thank you for all the valuable insights and constructive comments. We appreciate reviewers for highlighting the strengths of our work:

* **Clear Motivation (y7XW, kJL2, Xuuh):** Reviewers found the motivation of the work interesting (y7XW, kJL2) and reasonable (Xuuh, kJL2). They praised the focus on addressing instance-level and intra-class variations overlooked by prior methods (Xuuh) and considered the use of Wasserstein Distance reasonable (y7XW).
* **Methodological Soundness (y7XW, Xuuh):** Reviewers appreciated the detailed analysis and decomposition of the OT distance into three components (Xuuh) and valued the clear theoretical explanation in Section 4.2, where the methods are summarized through an approximated inequality (y7XW).
* **Strong Empirical Results and Efficiency (kJL2, i9LC, Xuuh):** The paper’s performance was consistently noted, with reviewers highlighting that it outperforms baselines (i9LC), shows **good improvements over previous methods**  (kJL2) and includes code enhancing result reliability (kJL2). It is also supported by extensive experiments (Xuuh), and its **negligible extra computation cost** was also praised (Xuuh).
* **Clarity and Writing (kJL2, Xuuh):** The paper was described as well-written (kJL2) and clear and easy to understand (Xuuh).

In response to the valuable feedback, we highlight below the key revisions and clarifications:

* **Expanded Experiments and Ablations:**
    * Conducted low-IPC experiments (IPC=1, 2, 5) on ImageNet-1K, achieving **3–6% absolute gains** over the best prior methods.
    * Added a direct comparison with WMDD (ICCV 2025), surpassing it by **up to 14.7%** on ImageNet-1K.
    * Performed a new ablation study on our OTM, showing its superiority over KL divergence, MMD, and the SelKD loss.
* **Enhanced Clarity and Visualization:**
    * Clarified key technical details, including our region-level soft label strategy, notation, and the derivation of Eq. (8).
    * Added **two new diagrams** to visually explain the motivation and causal mechanism of our LIA module.
    * Provided a quantitative diversity check to empirically support our LIA approach.
* **New Parameter Analysis:**
    * Included a sensitivity analysis for the number of Sinkhorn iterations $T$.

We are pleased that all reviewers initially gave positive ratings and that our rebuttal addressed their concerns. We will incorporate all new experiments and analyses into the final version.

---

### Decision · Program_Chairs · 2025-09-17

**Decision:**

Accept (poster)

**Comment:**

This paper proposes an optimal transport-based framework for generative dataset distillation, aligning distributional geometry between synthetic and real data. The method is well-motivated, technically sound, and empirically validated across multiple benchmarks, showing competitive or superior performance compared to prior dataset distillation approaches. Reviewers agreed that the work is original and based on resonable motivations. Main concerns included generalizability to additional experiment settings (low-IPC), sensitivity to OT hyperparameters, and the need to better distinguish the contribution from the related work “Dataset Distillation via the Wasserstein Metric” (arXiv:2311.18531, 2023). The rebuttal provided satisfactory clarifications on these points. Overall, the paper represents a novel, technically solid, and practically promising contribution to generative dataset distillation, and is recommended for acceptance.